# Interactome profiling of Crimean-Congo hemorrhagic fever virus glycoproteins

Shiyu Dai ®[1,2,6], Yuan-Qin Min[1,3,6], Qi Li[1,4], Kuan Feng[1,3], Zhenyu Jiang[1,4], Zhiying Wang[1], Cunhuan Zhang[1], Fuli Ren[1], Yaohui Fang[1,4], Jingyuan Zhang[1,4], Qiong Zhu[1,3], Manli Wang ®[1,3], Hualin Wang ®[1,3] ✉, Fei Deng ®[1,3] ✉ & Yun-Jia Ning ®[1,3,5] ✉

Crimean-Congo hemorrhagic fever virus (CCHFV) is a biosafety level-4 pathogen requiring urgent research and development efforts. The glycoproteins of CCHFV, Gn and Gc, are considered to play multiple roles in the viral life cycle by interactions with host cells; however, these interactions remain largely unclear to date. Here, we analyzed the cellular interactomes of CCHFV glycoproteins and identified 45 host proteins as high-confidence Gn/Gc interactors. These host molecules are involved in multiple cellular biological processes potentially associated with the physiological actions of the viral glycoproteins. Then, we elucidated the role of a representative cellular protein, HAX1. HAX1 interacts with Gn by its C-terminus, while its N-terminal region leads to mitochondrial localization. By the strong interaction, HAX1 sequestrates Gn to mitochondria, thus depriving Gn of its normal Golgi localization that is required for functional glycoprotein-mediated progeny virion packaging. Consistently, the inhibitory activity of HAX1 against viral packaging and hence propagation was further elucidated in the contexts of pseudotyped and authentic CCHFV infections in cellular and animal models. Together, the findings provide a systematic CCHFV Gn/Gc-cell protein-protein interaction map, but also unravel a HAX1/mitochondrion-associated host antiviral mechanism, which may facilitate further studies on CCHFV biology and therapeutic approaches.

Crimean-Congo hemorrhagic fever virus (CCHFV) is a widespread arthropod-borne virus, circulating in more than 50 countries of Asia, Africa, southern Europe, and the Middle East[1,2]. As a biosafety level-4 pathogen, CCHFV can cause a severe acute infectious disease in human, named Crimean-Congo hemorrhagic fever (CCHF), which is characterized by a rapid onset high-grade fever, gastrointestinal symptoms, thrombocytopenia, leukopenia, elevated liver enzymes, hemorrhage, multiple organ failure, and shock, with case fatality rates of up to 60%[3]. Due to the high virulence, multiple transmission routes and expanding geographical distribution, CCHFV poses a significant threat to public health and has been included in the World Health Organization (WHO) list of the top priority pathogens needing urgent research and development[4]. However, the requirement of high containment biosafety level facilities for experimental operation of CCHFV has long been a principal obstacle hindering research of the viral biology and development of antiviral therapies. Currently, there are no

[1]Key Laboratory of Virology and Biosafety and National Virus Resource Center, Wuhan Institute of Virology, Chinese Academy of Sciences, Wuhan 430071/430207, China. [2]Department of Cardiovascular Surgery of the First Affiliated Hospital & Institute for Cardiovascular Science, Suzhou Medical College, Soochow University, Suzhou 215006, China. [3]State Key Laboratory of Virology and Center for Biosafety Mega-Science, Chinese Academy of Sciences, Wuhan 430071/430207, China. [4]University of Chinese Academy of Sciences, 101408 Beijing, China. [5]Hubei Jiangxia Laboratory, Wuhan 430200, China. [6]These authors contributed equally: Shiyu Dai, Yuan-Qin Min. ✉e-mail: h.wang@wh.iov.cn; df@wh.iov.cn; nyj@wh.iov.cn

specific drugs or vaccines available[5]. Moreover, the knowledge of virus-host interactions including the mechanisms underlying viral infection or host response is quite limited, further impeding identification of therapeutic targets and design of specific anti-CCHFV strategies.

CCHFV belongs to the *Orthonairovirus* genus (*Nairoviridae* family, *Bunyavirales* order). Like all orthonairoviruses, CCHFV is an enveloped virus with a tripartite, single-stranded, negative-sense RNA genome comprising small (S), middle (M), and large (L) segments. These segments, S, M, and L, respectively encode the nucleoprotein (NP), glycoproteins (GPs), and RNA-dependent RNA polymerase (RdRp). In an infectious virion, the viral genomic segments encapsidated by NP in the ribonucleoprotein (RNP) complexes are surrounded by a host (more precisely Golgi apparatus)-derived lipid bilayer studded with spikes of structural GPs, Gn and Gc.

The GPs play crucial roles in multiple events of the virus life cycle including entry (receptor binding and membrane fusion), assembly (packaging, budding, and hence progeny propagation), tropism, and pathogenicity. CCHFV infection is initiated by the spike GP binding to cell surface receptor(s) unknown, followed by internalization of virions in clathrin-dependent endocytosis. As with other bunyaviruses, Gc of CCHFV is a putative class II membrane fusion protein[6,7], and Gn is predicted to be responsible for receptor binding. After passing through early endosomes to multivesicular bodies (MVBs) where CCHFV envelope fuses with the host membrane[8], viral RNPs are released in the cytoplasm, leading to subsequent replication and transcription of viral genomes and expression of viral proteins. Therein, glycoprotein production of nairoviruses is much more complicated than that of other bunyaviruses in the aspects of processing, size, and species of mature proteins. Upon initial protein synthesis, GPs undergo processing, transport, and maturation in the cellular endoplasmic reticulum (ER)-Golgi secretory pathway. First, the M segment expresses a polyprotein precursor (GPC) which is co-translationally cleaved by ER signal peptidases into a Gn precursor (PreGn), a Gc precursor (PreGc), and a nonstructural protein NSm[9,10]. The Gn and Gc precursors are subsequently processed by host proteases to generate the mature Gn, Gc, and several nonstructural secreted glycoproteins (NSGs) released from the PreGn N-terminus, including an O-glycosylated mucin-like protein, GP38 and their potential complexes of unknown constituent ratio or mechanism. Gn is high efficiently localized and accumulated in the Golgi that is the compartment of virion packaging and budding, while Gc is predominantly localized in ER and requires association with Gn to enable its transport to the Golgi and further maturation[11]. Gn plays particularly important roles in viral assembly not only by assisting Golgi localization of Gc but also by mediating virion packaging of newly synthesized RNPs (i.e., production of progeny virions) in the Golgi compartment[12]. Although many details (e.g., the functions of nonstructural proteins) in GP processing remain elusive, it seems also important for GP production and localization to be expressed in the context of the full-length glycoprotein (i.e., GPC) gene harboring encoding regions of nonstructural proteins (especially GP38 and NSm)[10,13–15]. Finally, nascent progeny virions are transported via vesicles from the Golgi to cell membrane and released extracellularly.

The multiple functions of CCHFV GPs especially in terms of virus entry and assembly, together with their orchestrated production and accumulation processes from ER to Golgi required for progeny virion morphogenesis, should involve various complex interactions of GPs with host cell factors, of which however, little is known. We here determined the cellular interactomes of CCHFV Gn/Gc by affinity purification coupled with tandem mass spectrometry (AP-MS/MS). Together with bioinformatics and a series of experimental validations, we demonstrated that the high-confidence Gn/Gc-interacting host factors are mainly distributed in cytoplasmic membranous organelles (including ER, mitochondria, Golgi, and internalization/secretion transport vesicle systems) and plasma membrane and likely participate in transmembrane transport, metabolism, oxidative stress, protein folding and glycosylation, etc. These Gn/Gc interactor-involved molecular functions, cellular components, biological processes, interaction networks and pathways are well linked to the various putative physiological actions of the viral glycoproteins, presenting a global view of the complex interactions between CCHFV GPs and host cells. Further, we unraveled the role of a representative GP-interacting host factor, HCLS1-associated protein X-1 (HAX1), and showed a cellular antiviral mechanism by HAX1 sequestrating Gn to mitochondria and thus disrupting the Golgi location and progeny virion propagation.

## Results
### Identification of host proteins interacting with CCHFV GPs
To better understand the molecular underpinnings of CCHFV infection, we performed AP-MS/MS experiments to define the host proteins that physically interact with CCHFV glycoproteins Gn and Gc in human embryonic kidney 293T (HEK293T) cells. HEK293T cells were chosen for proteomic study for several reasons: (1) they are permissive to CCHFV infection; (2) they are a well-recognized cell model derived from human kidney that is one of CCHFV-targeted organs clinically; (3) their ability to be efficiently transfected; and (4) the availability of annotated human proteome databases to facilitate protein identification and function assignment. As shown in Fig. 1a, we adopted a strategy to tag Gn or Gc with a twin-Strep-tag (~3 kDa) in the expression context of the full-length GPC, which enables highly efficient and specific purification of corresponding Strep-tag fusion proteins and their interacting partners from cell lysates[16]. Western blot analysis of cells transfected with the expression plasmids showed that the viral proteins were successfully expressed in the context of CCHFV GPC and high efficiently processed into Gn and Gc (Fig. 1b, c), consistent with previous observations[9,17]. Of note, no noticeable influence of the GPC expression on cell viability was observed in the experiments, which was further confirmed by CCK8 assay (Supplementary Fig. S1). Strep-tag affinity purification was then performed with the cell lysates and purified products were subjected to silver staining and additional Western blot analysis, respectively. As shown in Fig. 1d–g, in addition to Strep-tagged Gn or Gc (i.e., the baits), many specific bands were observed in the cells expressing GPC compared to control, suggesting successful co-precipitation of Gn/Gc-associated host proteins. Simultaneously, a small amount of Gn or Gc proteins were pulled down by each other (Fig. 1f, g), consistent with the putative interaction of a fraction of Gn and Gc in the context of CCHFV infection that enables Gc localization and packaging into virions in the Golgi. Furthermore, the affinity purification products from four biological replicates were subjected to quantitative MS analysis. Only proteins meeting the criteria, $p$ value < 0.05 and fold change (FC) > 2 in comparison with control, were here considered as Gn/Gc interactors of significant confidence. According to the criteria, a total of 34 (Supplementary Table S1) and 33 host proteins (Supplementary Table S2) were identified in the Strep-tagged Gn and Gc precipitates, respectively and therein, 22 host proteins were overlapped among them (Fig. 2a–c). These proteins likely represent direct or indirect host interactors or interacting complexes with Gn and/or Gc. Additionally, the nonstructural proteins, mucin and GP38 (but not NSm), were also identified as Gn and Gc interactors (Fig. 2a, b), consistent with their potential assisted roles in the viral glycoprotein processing as previously reported[10,13–15].

To further validate our proteomic data, CCHFV Gn/Gc interactions with several representative host proteins identified by MS was verified via independent Western blot analysis using specific antibodies. These host proteins included KPNB1, HAX1, THBS4, ATP1A1, RPN1, and HADHA. As shown in Fig. 2d, all the analyzed host molecules were confirmed in the pulldown products with CCHFV glycoproteins

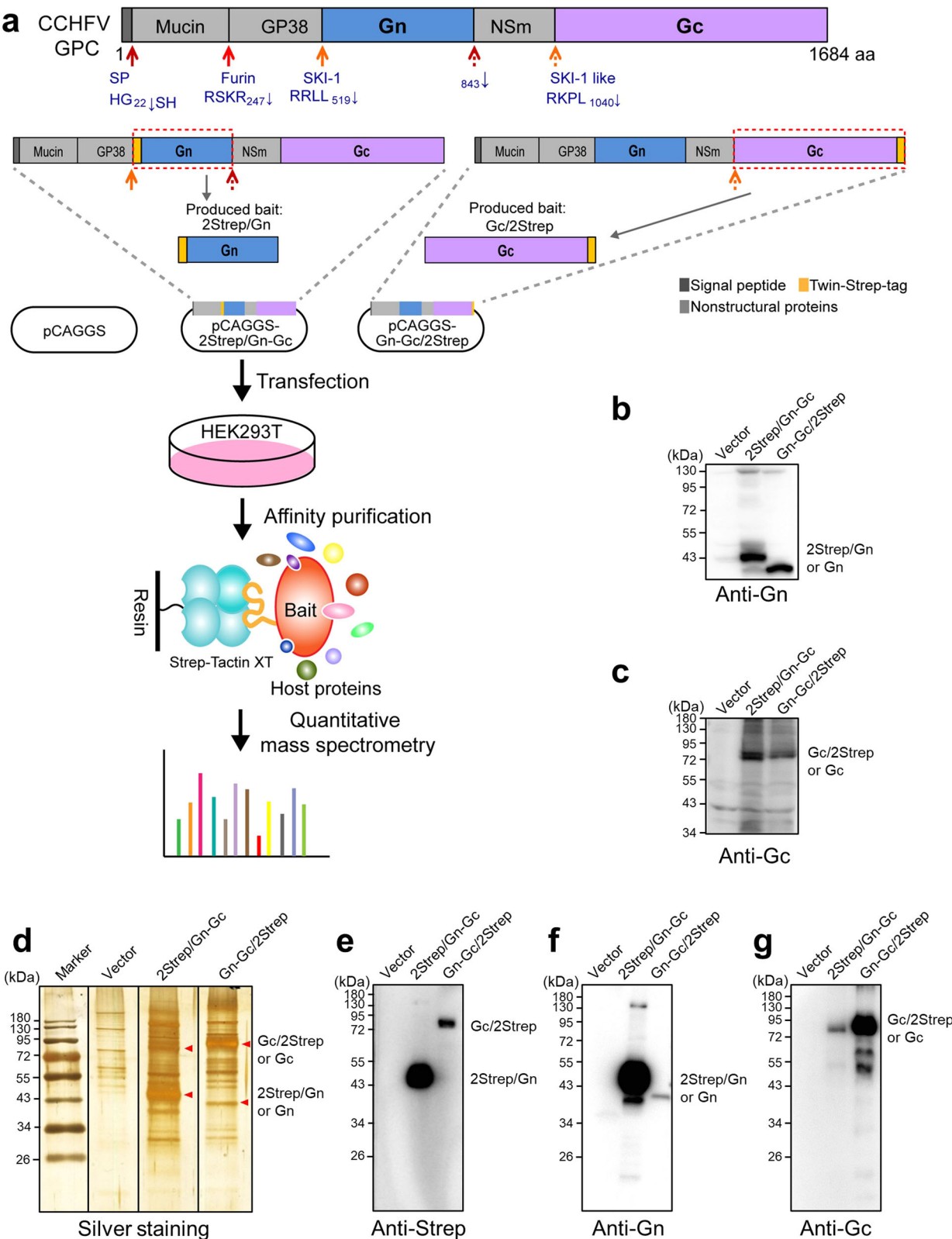

as baits in HEK293T cells. Although they were detected in both Gn and Gc co-precipitates, their band intensities in Western blot were overall in accord with the respective abundance detected by tandem MS, supporting the proteomic data. Furthermore, similar results were also reproduced by pulldown and Western blot using hepatic Huh7 cells (Fig. 2e), another CCHFV permissive cell model, validating the MS analyses as well.

## Bioinformatics analyses of CCHFV Gn/Gc-interacting host proteins

For functional annotation, the identified Gn/Gc-interacting host proteins were subjected to gene ontology (GO) analysis. The top over-represented GO terms with ranked −log10 (p values) were highlighted in bubble chart. As shown in Fig. 3a–c, the proteins were clustered into groups based on the cellular component (CC), biological process (BP),

**Fig. 1 | Strategy for identification of host proteins interacting with CCHFV Gn/Gc by AP-MS/MS. a** Schematic diagram of affinity-purification mass spectrometry quantification approach to identify host proteins interacting with CCHFV Gn/Gc. CCHFV M segment genome encodes the structural glycoproteins Gn and Gc, as well as several nonstructural proteins (mucin, GP38, and NSm), which are generated by cellular protease-mediated cleavage at the indicated sites. To produce specifically tagged Gn or Gc as the baits, a twin-Strep-tag was fused to the N-terminus of Gn after the SKI-1 cleavage site (i.e., RRLL$_{519}$) or the C-terminus of Gc, respectively, in the context of the full-length GPC. The plasmids encoding the indicated baits, Strep-tagged Gn (2Strep/Gn) or Gc (Gc/2Strep), were constructed based on pCAGGS and respectively named pCAGGS-2Strep/Gn-Gc and pCAGGS-Gn-Gc/2Strep. For affinity purification-mass spectrometry analyses, HEK293T cells were transfected with the corresponding bait expression plasmids or the control vector pCAGGS. Thirty-six hours post-transfection (h p.t.), cells were lysed and subjected to affinity purification by magnetic beads coated with Strep-Tactin. Purified proteins were digested into peptides by trypsin and subjected to quantitative mass spectrometry analysis. Four independent biological replicates (i.e., *n* = 4) were performed for each Strep-tagged protein or the control in AP-MS analysis. The cleavage sites along with several known cellular enzymes are indicated. SP signal peptidase, Furin furin protease, SKI-1 subtilisin kexin isozyme-1/site-1 protease, SKI-1 like a protease with similar specificity to SKI-1. **b**, **c** Western blot analysis showing the expression and cleavage of CCHFV GPC. The tagged and untagged Gn and Gc in the lysates were detected by anti-Gn (**b**) and anti-Gc (**c**) antibodies, respectively. GPC was efficiently processed into Gn and Gc, although a small fraction of preGn could also be detected (~140 kDa). **d** Silver staining of the Strep affinity purification products. Red arrows denote Gn or Gc in each lane. Western blot analysis was also conducted to show the bands of CCHFV Gn and Gc in the AP products using anti-Strep (**e**), anti-Gn (**f**), and anti-Gc (**g**) antibodies, respectively. The experiments were repeated for three times with similar results (**b**–**g**). Source data are provided as a Source Data file.

and molecular function (MF). GO CC term analysis of the 45 Gn/Gc-interactors demonstrated a strong enrichment of membrane-associated proteins, such as organelle membrane, mitochondrion, ER, extracellular exosome and cytoplasmic vesicle (Fig. 3a, d). The biological function analysis showed that proteins responsible for transmembrane transport and ATP metabolic process were highly overrepresented, comprising the proteins COX5A, MT-CO2, NDUFA4, ATP1A1, ATP2A2, ATP5A1, ATP5B, ATP5D, ATP5H, HSPA8, and HSPA1B (Fig. 3b, d). In addition, molecules involved in the protein folding and protein binding processes were also significantly enriched in the dataset, hinting at possible roles of these proteins in CCHFV GP physiology.

To enable a more detailed analysis of the interactions covered by our AP-MS/MS approach, protein networks were performed by STRING algorithm. Accordingly, 97 high-confidence protein interactions were identified (Fig. 3e–g). In this protein-protein interaction network, there were two clusters recognized as notable subnetworks based on the interaction numbers. One was composed of ATP5B, MT-CO2, ATP5H, MT-CO2, COX5A, ATP5D, ATP5A1, and NDUFA4. The other consisted of HSPA5, HSPA8, PHB, HSPB1, TUBB4B, TUBB2A, TUBA1C, and TUBB. These "core" networks encompassed the most highly interconnected preys and were enriched in mitochondrion, ER, and cytoskeleton, suggesting that these host factors may play a key role in the GP-host interaction network.

Furthermore, we mapped the host proteins to CCHFV life cycle, based on the GO and interaction network analyses. A potential global view of CCHFV Gn/Gc interactions with host cells was sketched out (Fig. 3h). These host factors mainly locating on the ER-Golgi system, vesicles, mitochondria, cytosol, and plasma membrane could be involved in the protein processing and transport, vesicle trafficking, cytoskeleton-mediated intracellular transport, energy metabolism, cell adhesion, etc. The findings seem well consistent with the supposed physiological activities of viral glycoproteins during virus entry, their production and transport, and virion morphogenesis along membranous organelles (Fig. 3h). By these active or passive interactions, the viral proteins may regulate or be regulated by the cellular factors to promote virus infection and pathogenicity or in turn lead to host antiviral defense, meriting further investigations.

### Interaction and colocalization of HAX1 with CCHFV glycoproteins

Among these interactors, HAX1, which exhibits significant interaction with Gn, followed by Gc (Supplementary Tables S1 and S2 and Fig. 2d, e), predominantly locates on mitochondria[18]. The glycoproteins of CCHFV as a representative bunyavirus, however, are generated and processed in the ER and Golgi complex. Moreover, there is no report of HAX1 targeting of bunyaviruses. We asked what the biological implications of the interaction between CCHFV glycoproteins and this

mitochondrial protein could be in the viral infection. Thus, in this study, we next investigated the role of HAX1 as a potential bunyavirus-targeting host factor.

Firstly, aside from the validation by Strep-AP assay, the association of HAX1 with CCHFV Gn/Gc was also detected by a reciprocal interaction analysis via S-pulldown assay with HAX1 as bait. The results demonstrated that Gn, superior to Gc, indeed can be notably coprecipitated by HAX1 (Fig. 4a), confirming the protein interactions and supporting the stronger binding of HAX1 with Gn as seen above (Fig. 2d, e and Supplementary Tables S1 and S2). Then, cellular distribution of CCHFV Gn/Gc and HAX1 was monitored by immunofluorescence assay (IFA) and confocal microscopy. As shown in Fig. 4b, in cells expressing GPC alone, Gn mainly accumulated as patches in perinuclear regions of the cells indicative of Golgi apparatus localization, whereas Gc distributed more diffusely around nuclei in agreement with its predominant ER localization as previously observed[11,19]. Further, in GPC and HAX1 co-expressing cells, Gn was notably colocalized with HAX1, while colocalization of Gc with HAX1 was observed but to a lesser extent in comparison (Fig. 4b, c). More intriguingly, a dramatic change of Gn localization from perinuclear aggregation to dispersive distribution accompanying with overexpressed HAX1 was observed (Fig. 4b), which not only further supports the results of protein interactions but also reflects a potential effect of HAX1 on CCHFV glycoproteins and in particular Gn.

### HAX1 hijacks the viral glycoproteins (especially Gn) to mitochondria, disrupting the Golgi apparatus localization

Knowing that localization of CCHFV Gn to the Golgi apparatus is necessary for viral packaging, we further investigated whether HAX1 interferes with Golgi apparatus localization of Gn. The following IFAs showed that in absence of HAX1 overexpression, Gn exhibits evident Golgi apparatus distribution as indicated by its efficient colocalization with the Golgi marker (B4GALT1) (Fig. 5a). However, in cells co-expressing HAX1, the localization pattern of Gn was obviously changed from Golgi apparatus patches to dispersed cytoplasmic distribution (Fig. 5a), consistent with the aforementioned observation in Fig. 4b. Together with Pearson's correlation coefficient (PCC) analyses, it was clearly demonstrated that consistent with the potent HAX1-Gn interaction, Gn is strongly colocalized with HAX1 and meanwhile the proper Golgi localization of Gn is dramatically deprived by HAX1 (Fig. 5a, c). These results corroborate that HAX1 likely disrupts the Golgi localization of Gn by its robust interaction and colocalization with the viral protein.

As HAX1 was previously reported to locate primarily on mitochondria, we next examined mitochondrial localization of the host and viral proteins by additional IFAs using mitochondrial marker (MitoTracker)[20]. Consistently, the dramatic change of Gn localization

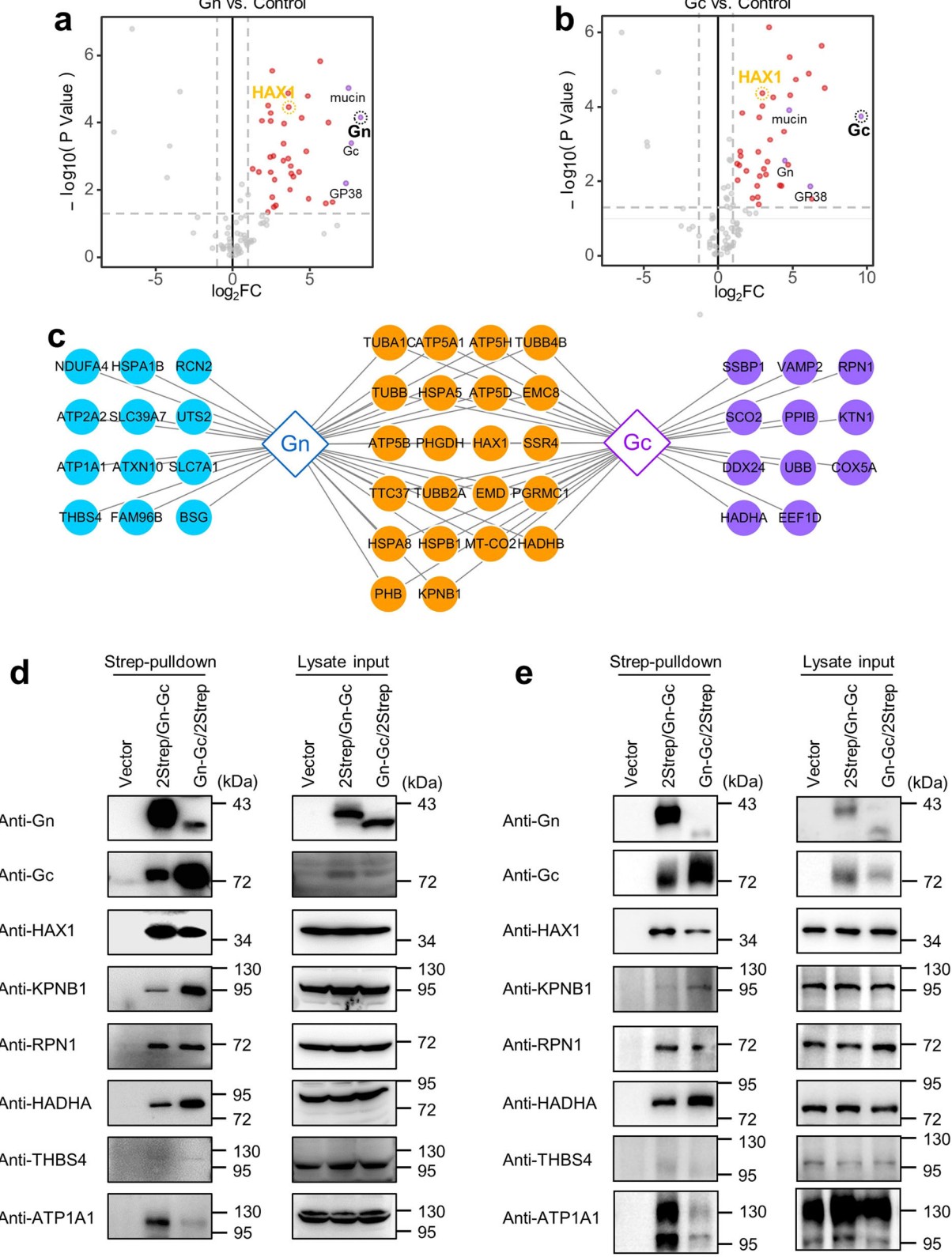

**Fig. 2 | Identification and validation of CCHFV Gn/Gc interacting host proteins.** Volcano plots of proteins identified in CCHFV Gn (**a**) or Gc (**b**) affinity purification mass spectrometry. The quantitative fold change (FC) of each protein was compared to control and *p* value was determined by Student's *t* test (two-tailed distribution, two-sample, equal variance). CCHFV Gn or Gc interacting host proteins (log2FC > 1, *p* value < 0.05) were labeled in red and the viral proteins were labeled in purple. **c** Interacting host proteins enriched by CCHFV Gn (blue), Gc (purple) or both Gn and Gc (orange). **d**, **e** Validation of the interactions between CCHFV Gn/Gc and the identified representative host proteins by Western blot analysis. HEK293T (**d**) or Huh7 (**e**) cells were respectively transfected with pCAGGS-Gn-Gc/2Strep, pCAGGS-2Strep/Gn-Gc, or the pCAGGS vector as a control. At 36 h p.t., cells were lysed for Strep-pulldown assays. The cell lysates (lysate input) and Strep-tag pulldown products (Strep-pulldown) were analyzed by Western blot analysis using indicated antibodies, respectively. These experiments were repeated for three times with similar results (**d**, **e**). See also Supplementary Tables S1 and S2. Source data are provided as a Source Data file.

pattern by HAX1, colocalization of Gn with HAX1, and mitochondrial distribution of HAX1 were all repeatedly observed. Moreover, interestingly, we found that Gn is indeed significantly relocated by HAX1 to mitochondria (Fig. 5b, d). A similar analysis of Gc further showed that Gc also seems to be hijacked by HAX1 to mitochondria, but to less extent (Supplementary Fig. S2). Additionally, partial localization of Gn (as well as Gc) on mitochondria could also be observed in the context of only endogenous HAX1 expression (Supplementary Fig. S3), indicating the sequestration by endogenous HAX1. Together, these findings suggest that by the protein interactions, HAX1 can sequestrate the viral envelope proteins (especially Gn) to mitochondria, thus subverting the proper Golgi localization of the viral protein required for efficient virion packaging.

To further validate the hijacking of the viral proteins by HAX1 to mitochondria, we also conducted mitochondrial fractionation assays. These experiments were expected to be complementary and have the advantage of analyzing total cell population to enrich the signals, while it became difficult to note relatively smaller localization changes in extents by observing individual cells through IFA and confocal microscopy. As shown in Western blot analyses, along with the enrichment of HAX1, Gn was also notably enriched in the extracted mitochondrion samples even in the context of only endogenous HAX1 expression (Fig. 5e and Supplementary Fig. S5). Moreover, HAX1 overexpression further increased the mitochondrial localization of Gn, whereas HAX1 knockout (KO) by CRISPR-Cas9 gene editing dramatically ablated the enrichment of Gn on mitochondria (Fig. 5e and Supplementary Figs. S4 and S5). These data further substantiate HAX1 hijacking of Gn to mitochondria. Additionally, similar results were also obtained for Gc, but to less extents.

## Both the N- and C-termini of HAX1 are required for its hijacking of Gn to mitochondria

It was reported that HAX1 contains several putative characteristic regions including two N-terminal BH domains with limited homology with the BH1 and BH2 domains of Bcl-2 family members and a PEST (proline, glutamic acid, serine, threonine) motif[18,21]. However, these homologies or domains and their functional significance are not well experimentally documented and remain to be further determined[18]. To address the domain(s) of HAX1 required for the targeting of CCHFV Gn, we constructed several HAX1 truncated mutants based on the predicted linear organization (Fig. 6a). Pulldown assays showed that the mutant (HAX1-C1) consisting of the C-terminal 104–279 aa and another shorter mutant (HAX1-C2) containing the region of 118–279 aa both exhibit evident interaction with Gn, albeit to less extent compared to the wild-type HAX1 (Fig. 6b). However, the N-terminal 1–117 aa (HAX1-N) failed to be coprecipitated with Gn (Fig. 6b). The results suggest that HAX1 C-terminus is likely the binding region for Gn, although the N-terminal sequence may also have some supportive effect on the strong interaction of HAX1 with Gn. Furthermore, IFA and confocal microscopy analyses further supported and extended these findings. As shown in Fig. 6c, the C-terminal-deleting mutant HAX1-N was still mainly localized at mitochondria, but consistent with the above protein interaction analyses, failed to be colocalized with Gn. In comparison, the mutants HAX1-C1 and HAX1-C2, which lack the N-terminus, exhibited different distribution patterns with HAX1-N and the full-length HAX1 and lost the predominant mitochondrial localization (Fig. 6c). Moreover, like HAX1-N, they could not relocate Gn to mitochondria either, although they were still partially colocalized with Gn (Fig. 6c). Thus, both of the N- and C-termini are required for HAX1 hijacking of Gn to mitochondria, as the C-terminal region of HAX1 is needed for its interaction with Gn and the N-terminus of HAX1 is important for targeting to mitochondria. Collectively, these data further confirm HAX1 targeting of CCHFV Gn and suggest that HAX1 traps Gn to mitochondria through its C-terminal interaction with Gn and N-terminal targeting to mitochondria.

## HAX1 inhibits CCHFV GP-mediated virion packaging and acts as a significant host restriction factor of CCHFV

Little is known on host factors regulating CCHFV infection to date. CCHFV virions are packaged by the viral glycoproteins (particularly Gn) at the Golgi compartment and there obtain their envelope membrane in situ[12]. Given the sequestration of Gn by HAX1 to mitochondria, we speculated that the host factor HAX1 may have negative impact on the glycoprotein-mediated virion packaging and hence CCHFV propagation. Firstly, we assessed the effect of HAX1 on CCHFV glycoprotein activities using a GFP-expressing pseudotyped VSV (vesicular stomatitis virus) reporter system bearing CCHFV glycoproteins[22,23]. Due to deletion of the VSV glycoprotein gene, the pseudotype viruses need complement of glycoprotein expression by plasmid transfection for nascent progeny packaging; otherwise, the pseudotypes can only undergo a single cycle of infection. Their infectivity is dictated by the harbored heterologous glycoproteins. Therefore, the pseudotype system has been adopted as a classical tool for studying the entry or packaging processes mediated by viral envelope glycoproteins. As shown in the analysis of invasion to cells (Fig. 7a–d), HAX1 did not notably affect the infection efficiency of the pseudovirus harboring CCHFV glycoproteins (VSV△G/GFP-CG) that was prepared in advance, suggesting HAX1 likely does not affect CCHFV glycoprotein-directed invasion. However, interestingly, the packaging efficiency of VSV△G/GFP-CG indeed significantly decreased when it was produced in the presence of HAX1 over-expression (Fig. 7e–h). Additionally, in comparison, neither packaging nor invasion of the pseudovirus bearing VSV glycoprotein (VSV△G/GFP-VG) was significantly affected by HAX1 (Supplementary Fig. S6), manifesting the specific interference of HAX1 with CCHFV glycoprotein-driven virion packaging. Together, these results indicate that by trapping Gn to mitochondria, HAX1 inhibits the glycoprotein-mediated packaging of progeny virions.

Based on the data obtained above, we considered that HAX1 could be a notable host restriction factor against CCHFV infection. Interestingly, HAX1 expression in host cells was significantly upregulated in response to CCHFV infection even at low doses (MOI = 0.1 or 0.3) (Fig. 8a). Next, we analyzed the effect of HAX1 on CCHFV infection by gain/loss-of-function assays. Firstly, the overexpression of HAX1 indeed evidently inhibits CCHFV infection (Fig. 8b–e). Notably, HAX1 overexpression was confirmed to substantially reduce propagation of progeny CCHFV virions, in accordance with the uncovered activity of HAX1 to inhibit nascent virion packaging (Fig. 8d, e). Furthermore, the anti-CCHFV activity of HAX1 was also revealed by the notable increase of viral protein and RNA abundances in cells with HAX1 knockdown (KD) by RNAi or KO by CRISPR-Cas9 gene editing (Fig. 8f–h and Supplementary Fig. S4). Moreover, consistently, analyses of viral loads and virus titration showed that production of newly packaged progeny virions from HAX1-KO cells is significantly enhanced (Fig. 8i, j), further confirming the role of HAX1 as a host restriction factor against CCHFV multiplication.

Since HAX1 exhibits significant anti-CCHFV activity in vitro, we then asked whether HAX1 also has an impact on virus infection in vivo. Due to the necessary roles in B-cell and nerve system development, HAX1 knockout in mouse results in premature death[24]. Thus, we established a HAX1-knockdown mouse model by transient transduction of a lentiviral vector expressing specific shRNA as previously described[25]. Despite no severe diseases or death, adult immunocompetent mice could be infected by CCHFV with detectable virus in multiple tissues[26,27]. Prior to infection, 6–8-week-old female C57BL/6 mice were transduced with the HAX1-targeting or control shRNA expression vectors. One week post transduction, mice were infected with CCHFV. Animals were monitored daily for clinical manifestations of disease and sacrificed for organ collection to analyze viral infection at 3 d p.i. No severe clinical signs or deaths were observed in all infected animals.

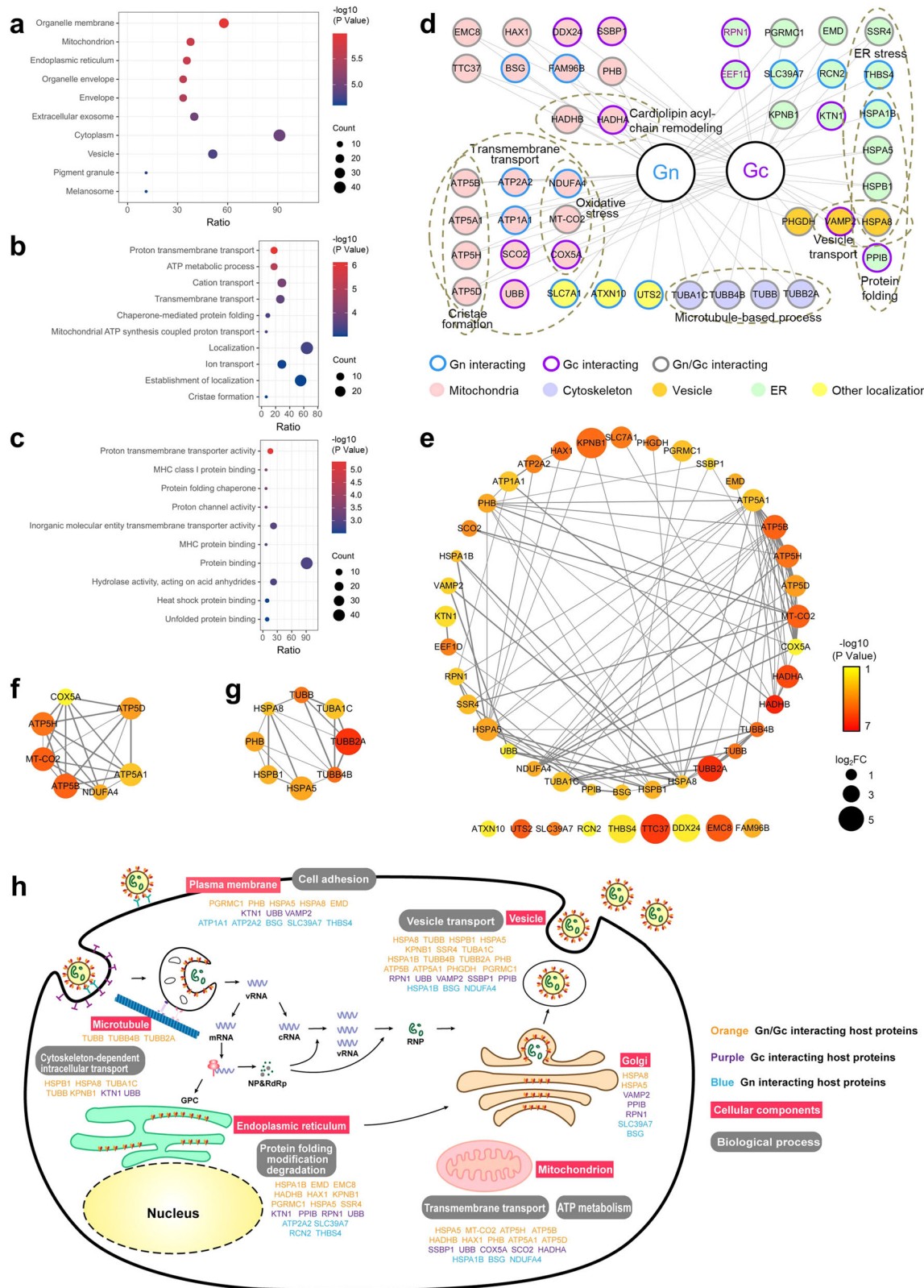

Although HAX1 was only partially silenced by the specific shRNA in tested mouse organs (Fig. 8k), viral loads in the organs of mice with HAX1 KD were increased compared to the controls (Fig. 8l), indicating that knockdown of HAX1 promotes CCHFV propagation in vivo. These data further corroborate the role of HAX1 as a host restriction factor against CCHFV.

## Discussion

CCHFV is one of the most dangerous human pathogens known to date and a potential biological agent of bioterrorism, posing a serious threat to public health[4]. Nonetheless, the requirement for high biosafety level containment, together with the limitations of experimental materials and models for CCHFV study, have greatly hindered the

**Fig. 3 | Bioinformatics analyses of CCHFV Gn/Gc interacting host proteins.**
**a**–**c** Gene Ontology (GO) enrichment analysis was performed on CCHFV Gn/Gc interacting host proteins. The ID list of 45 high-confidence CCHFV Gn/Gc host interactors was submitted to PANTHER for GO analysis based on their roles in cellular component (**a**), biological process (**b**), and molecular function (**c**). The top 10 GO terms were selected for visualization. Fisher's exact test with False Discovery Rate (FDR) correction was performed. **d** Overview of function and localization enrichment of CCHFV Gn/Gc interacting host proteins. CCHFV bait proteins and interacting host proteins are marked by frame with indicated colors. The sub-cellular localization of host proteins is labeled with the indicated colors. Proteins with known functions in transmembrane transport, oxidative stress, cardiolipin acyl-chain remodeling, cristae formation, microtubule-based process, vesicle transport, protein folding, or ER stress are circled in dashed border. Note that one protein may be classified into multiple subsets and only representative subsets were shown. **e** CCHFV Gn/Gc-interacting host factor protein-protein interaction network. The CCHFV Gn/Gc interacting host proteins were submitted to STRING for interaction network analysis. Size of the nodes is proportional to their fold change and only the larger fold change quantified in the two datasets is shown. Color of the nodes is related to their *p* values which were determined by paired two-tailed *t*-test. Edge thickness is relative to the strength of data support. **f**, **g** The "core" networks that were filtered through the MCODE plugin of Cytoscape. **h** A potential global view of CCHFV Gn/Gc cellular interactome in CCHFV life cycle. According to the results of the quantitative proteomic analysis, CCHFV Gn/Gc interactors were sorted and mapped to CCHFV life cycle based on their cellular localization and biological functions.

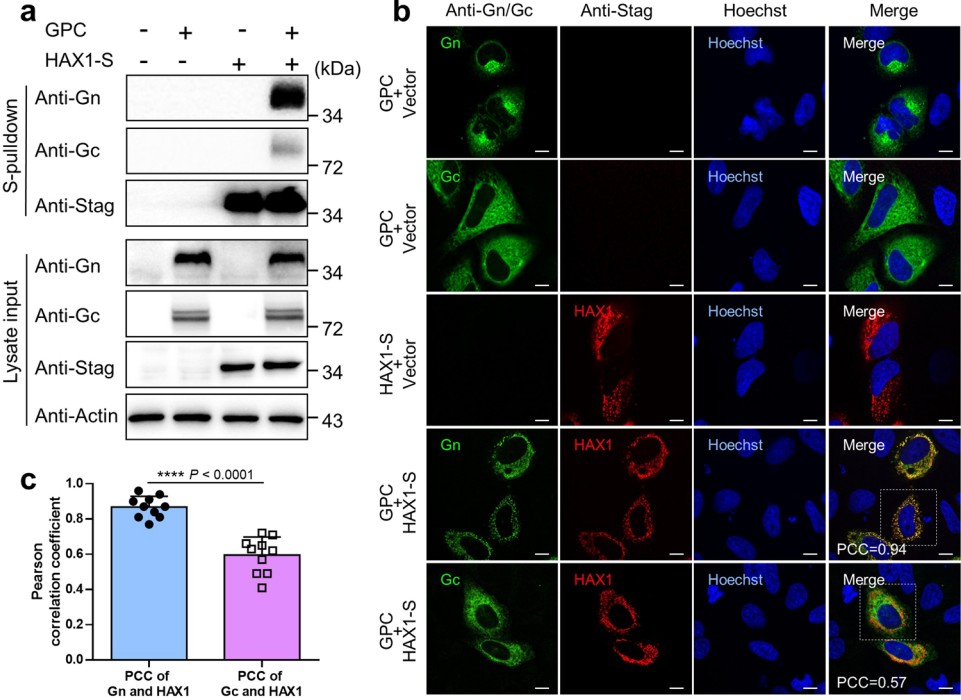

**Fig. 4 | Interaction and colocalization of HAX1 with CCHFV Gn/Gc. a** Validation of the interactions between CCHFV Gn/Gc and HAX1 by S-pulldown and Western blot analysis. HEK293T cells were transfected with plasmids encoding the indicated proteins or the control vector. At 36 h p.t., cells were lysed for S-pulldown assays, followed by Western blot analysis using indicated antibodies respectively.
**b** Subcellular localization and colocalization of CCHFV Gn/Gc and HAX1. HeLa cells were transfected with the indicated expression plasmids or control. After 36 h, cells were fixed, stained with anti-Gc, anti-Gn, or anti-Stag antibodies and visualized by confocal microscope. Nuclei stained with Hoechst are shown in blue. Pearson's correlation coefficients (PCC) for Gn/Gc and HAX1 in the representative cells in dashed box were indicated. Bars, 10 μm. **c** Co-localization analysis of Gn/Gc and HAX1 by PCC statistics. Colocalization between Gn/Gc and HAX1 was respectively assessed by PCC using the *coloc*2-plugin of the extended ImageJ version Fiji as described in "Materials and methods". Data are presented as means ± SD ($n = 10$ cells). Statistical significance in (**c**) was determined by two-tailed unpaired *t*-test. ****$p < 0.0001$. Data are representative of three independent experiments with similar results (**a**, **b**). Source data are provided as a Source Data file.

progress in molecular mechanisms of the viral infection and host interactions and hence the development of vaccines and drugs. The GPs of CCHFV are responsible for viral entry, tissue tropism, assembly (packaging/budding), and pathogenicity. However, it remains unclear which and how host factors are involved in these processes. Defining the proteome associated with the viral GPs not only can provide insights into infection and pathogenesis of CCHFV but also may reveal potential targets for design of host-directed interventions. In this study, we thus performed a comprehensive assessment of CCHFV Gn/Gc-interacting host factors based on AP-MS/MS strategy, establishing cellular interactomes of these pivotal viral components. Our interactome analysis reveals a broad array of host interplays for CCHFV glycoproteins in the virus life cycle (Fig. 3h) and may provide potential targets for rational design of antiviral drugs. Furthermore, we detailedly depicted the function and mechanism of HAX1, a representative protein in the interactomes identified here, by in vitro and in vivo experiments. Via its C-terminal interaction with Gn and N-terminal mitochondrial targeting, HAX1 acts as a host restriction factor against CCHFV by sequestering Gn to mitochondria to abate the viral glycoprotein-mediated packaging at the Golgi apparatus and hence progeny propagation (summarized in a proposed model, Fig. 9). These findings present an unexpected HAX1/mitochondrion-associated host antiviral strategy and also reflect the value of the interactomes established in this study.

HAX1 is a 35-kDa cytoplasmic protein that is predominantly localized on mitochondria by association with the cytosolic face of mitochondria[18]. Based on previous reports, HAX1 also is detectable in the ER and cytosolic vesicles, albeit to less extents, suggesting that HAX1 may have traffic or shuttling between these subcellular sites but high efficiently be concentrated to mitochondria[21,28,29]. Indeed, several

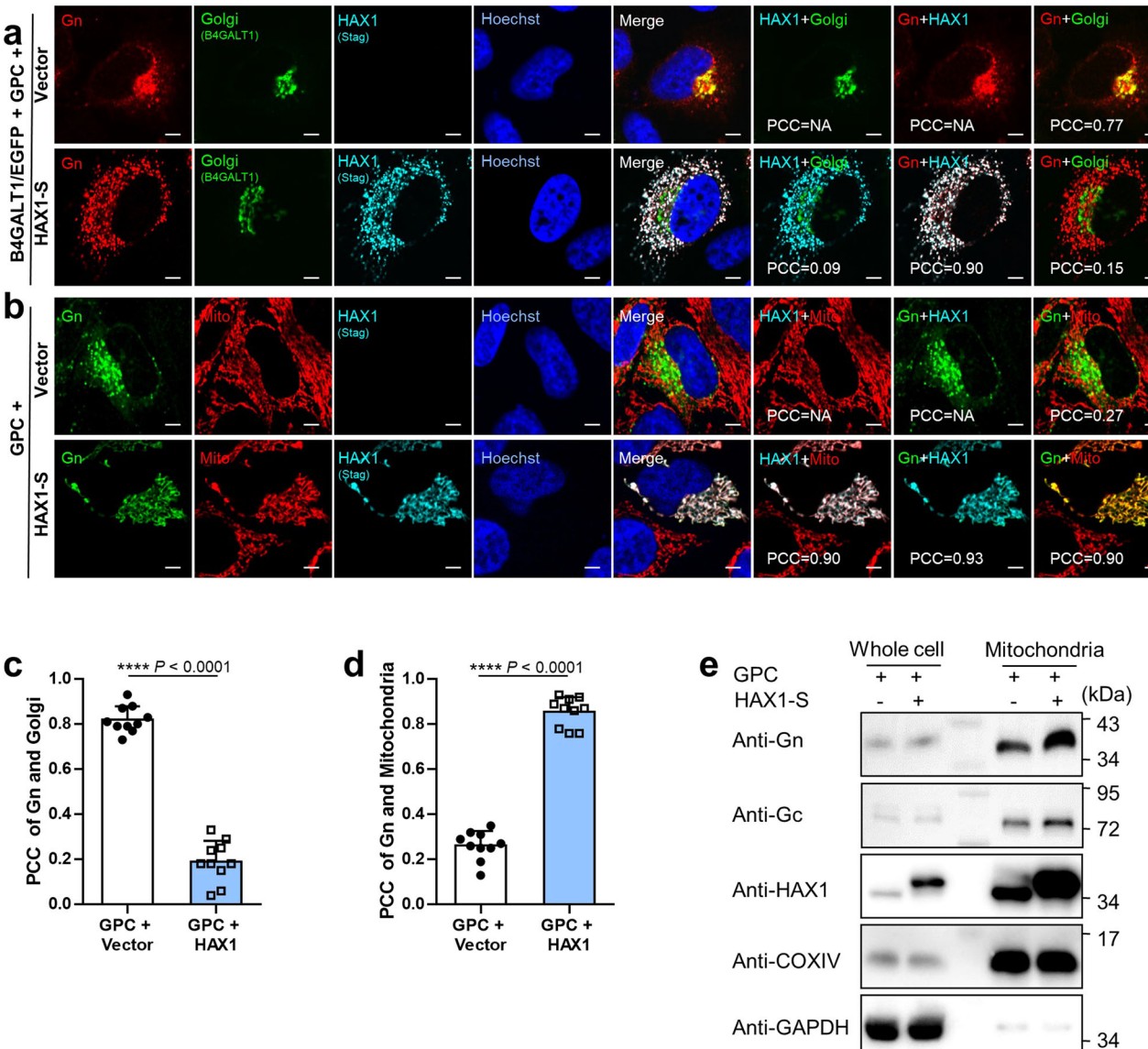

**Fig. 5 | HAX1 hijacks the viral glycoproteins (especially Gn) to mitochondria, reducing the Golgi compartment localization. a** Effect of HAX1 on Golgi localization of Gn. HeLa cells were transfected with plasmids encoding the indicated proteins, CCHFV GPC, B4GALT1 fused with EGFP (B4GALT1/EGFP, as the Golgi marker), HAX1 fused with Stag (HAX1-S), or the control vector. At 36 h p.t., cells were fixed and the localization of Gn (red), HAX1 (cyan), and B4GALT1/EGFP (green) were visualized by confocal microscopy after immunofluorescence staining. Nuclei stained with Hoechst are shown in blue. Representative images are shown. PCC Pearson's correlation coefficient. Bars, 5 μm. **b** Effect of HAX1 on mitochondrial localization of Gn. HeLa cells were transfected with plasmids encoding the indicated proteins or the control vector. At 36 h p.t., mitochondria were stained with MitoTracker red CMXRos prior to fixation, followed by detection of Gn (green), HAX1 (cyan), and mitochondria (red) by IFA and confocal microscopy. Bars, 5 μm. **c** Co-localization analysis of Gn and Golgi apparatus by PCC statistics.

Colocalization between Gn and the Golgi marker was assessed by PCC using the *coloc*2-plugin of the extended ImageJ version Fiji as described in "Materials and methods". Data are presented as means ± SD ($n = 10$ cells). **d** Co-localization analysis of Gn and mitochondria. Colocalization between Gn and mitochondria was assessed as in (**c**). Data are presented as means ± SD ($n = 10$ cells). ****$p < 0.0001$. **e** Mitochondrion fractionation assays showing Gn/Gc enrichments to mitochondria and their increases by HAX1 overexpression. HEK293 cells were transfected with plasmids encoding the indicated proteins or the control vector. At 36 h p.t., the mitochondrial fractions were isolated and subjected to Western blot analysis along with the whole-cell lysates. COXIV was used as the marker for mitochondria. See also Supplementary Figs. S2, S4 and S5. Statistical significance in (**c**, **d**) was determined by two-tailed unpaired *t*-test. Data are representative of three independent experiments with similar results (**a**, **b**, **e**). Source data are provided as a Source Data file.

studies have demonstrated HAX1 interaction with other cellular proteins of various subcellular distribution and its potential of or involvement in vesicular trafficking across cellular compartments[28,30–33]. However, the predominant mitochondrial localization of HAX1 suggests its dominantly stronger localization or final retention to mitochondria. In our study, we confirmed the major mitochondrial localization of HAX1 that is mediated by the N-terminus of the protein, consistent with previous reports[34]. Bunyaviruses, including CCHFV, assemble and bud from the Golgi apparatus and the viral Gn plays

important roles in the progeny virion morphogenesis in the Golgi apparatus. Gn itself locates in the Golgi, assists Gc to be transported to the Golgi, and there packages the viral RNPs into nascent virions[9,12]. We here further showed that HAX1 strongly interacts with Gn by its C-terminus and relocates Gn to mitochondria, disrupting the proper subcellular location of Gn for packaging and inhibiting CCHFV propagation. Despite the substantial Golgi localization, Gn is initially synthesized in the ER before its transport via the ER-Golgi trafficking pathway. As aforementioned, the main location of HAX1 is

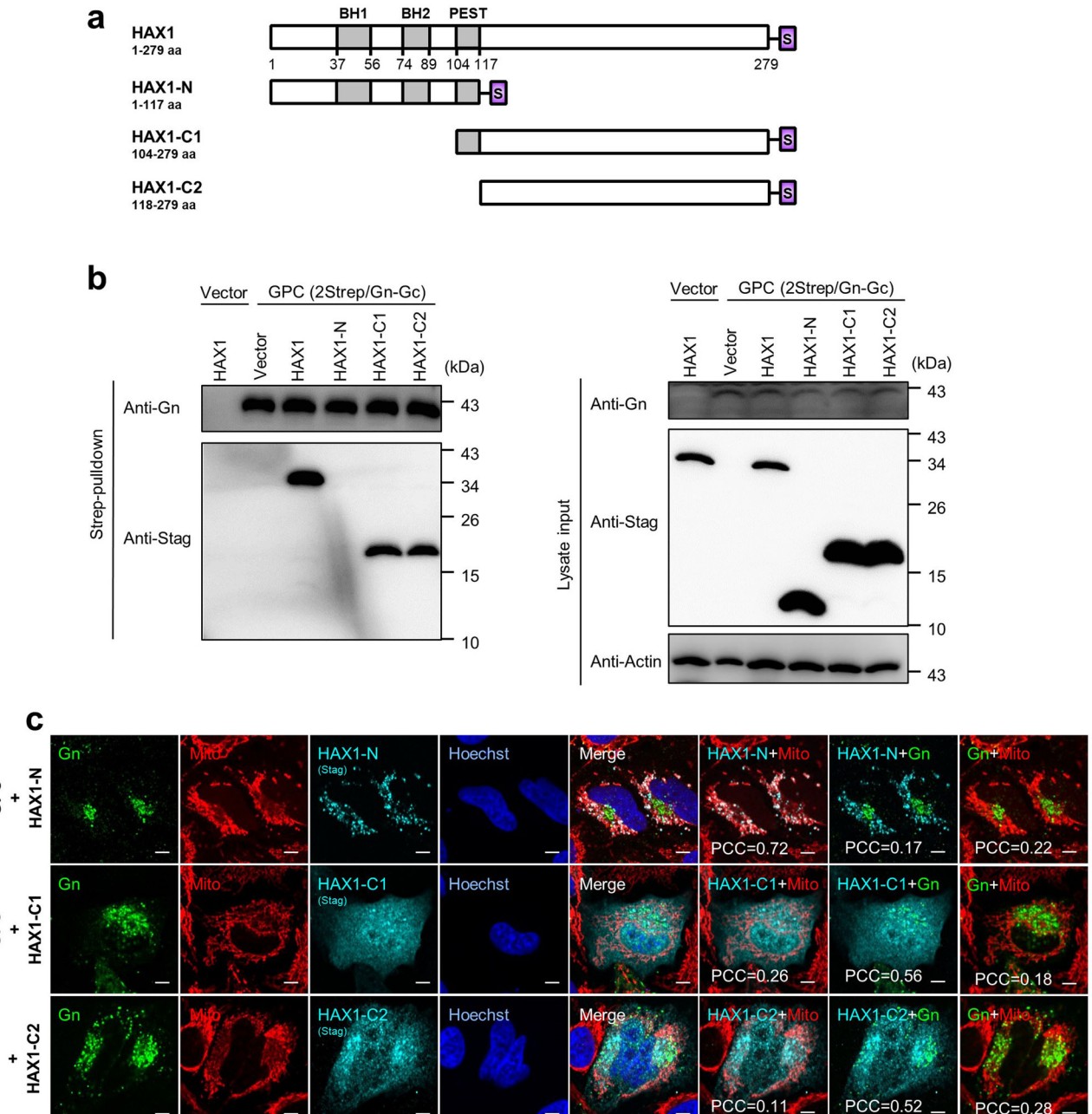

**Fig. 6 | C-terminus of HAX1 is required for interaction with Gn and N-terminus of HAX1 is essential for mitochondrial targeting. a** Schematic representation of full-length and truncated HAX1 fused with an Stag at the C-terminus. Predicted Bcl2 homology domains 1 and 2 (BH1 and BH2) and PEST (proline, glutamic acid, serine, threonine) sequence are indicated. **b** HEK293T cells were transfected with plasmids encoding the indicated proteins, CCHFV GPC (pCAGGS-2Strep/Gn-Gc), full-length or truncated HAX1 fused with Stag, or the control vector. At 36 h p.t., cells were lysed for Strep-pulldown assays, followed by Western blot analysis using indicated antibodies respectively. **c** HeLa cells were transfected with plasmids encoding the indicated proteins or the control vector. At 36 h p.t., mitochondria (red) were stained with MitoTracker red CMXRos, followed by detection of Gn (green) and HAX1 (cyan) by IFA. Nuclei stained with Hoechst are shown in blue. Representative images are shown. Bars, 5 μm. Colocalization between the indicated signals was assessed by Pearson's correlation coefficient as described in Fig. 5. These experiments were repeated for three times with similar results (**b**, **c**). Source data are provided as a Source Data file.

mitochondria, followed by ER[21,28]. Thus, it is very likely that HAX1 interacts with and hijacks Gn (as well as Gc to a lesser extent) from the sites (especially ER), where both HAX1 and Gn exist, to mitochondria. Future investigations of the biology of HAX1 (including its spatio-temporal characteristics) may help facilitate further understanding of its regulatory functions on the viral and cellular processes. Additionally, we also observed that HAX1 expression is upregulated in cells in response to CCHFV infection, which may be an add-on of the host cell antiviral strategy via HAX1, contributing to the antiviral effect of HAX1,

although the mechanism underlying the enhancement of HAX1 expression upon CCHFV infection needs to be further determined. Consequently, this study uncovers an unexpected antiviral mechanism of host cell, providing valuable insights into virus-host interactions and particularly host defense strategies.

HAX1 is ubiquitously expressed at a fairly high level in a wide range of tissues and seems to be involved in anti-apoptosis, regulation of cell motility and calcium homeostasis, cancer progression, and severe congenital neutropenias (SNC, a human genetic disease)[18].

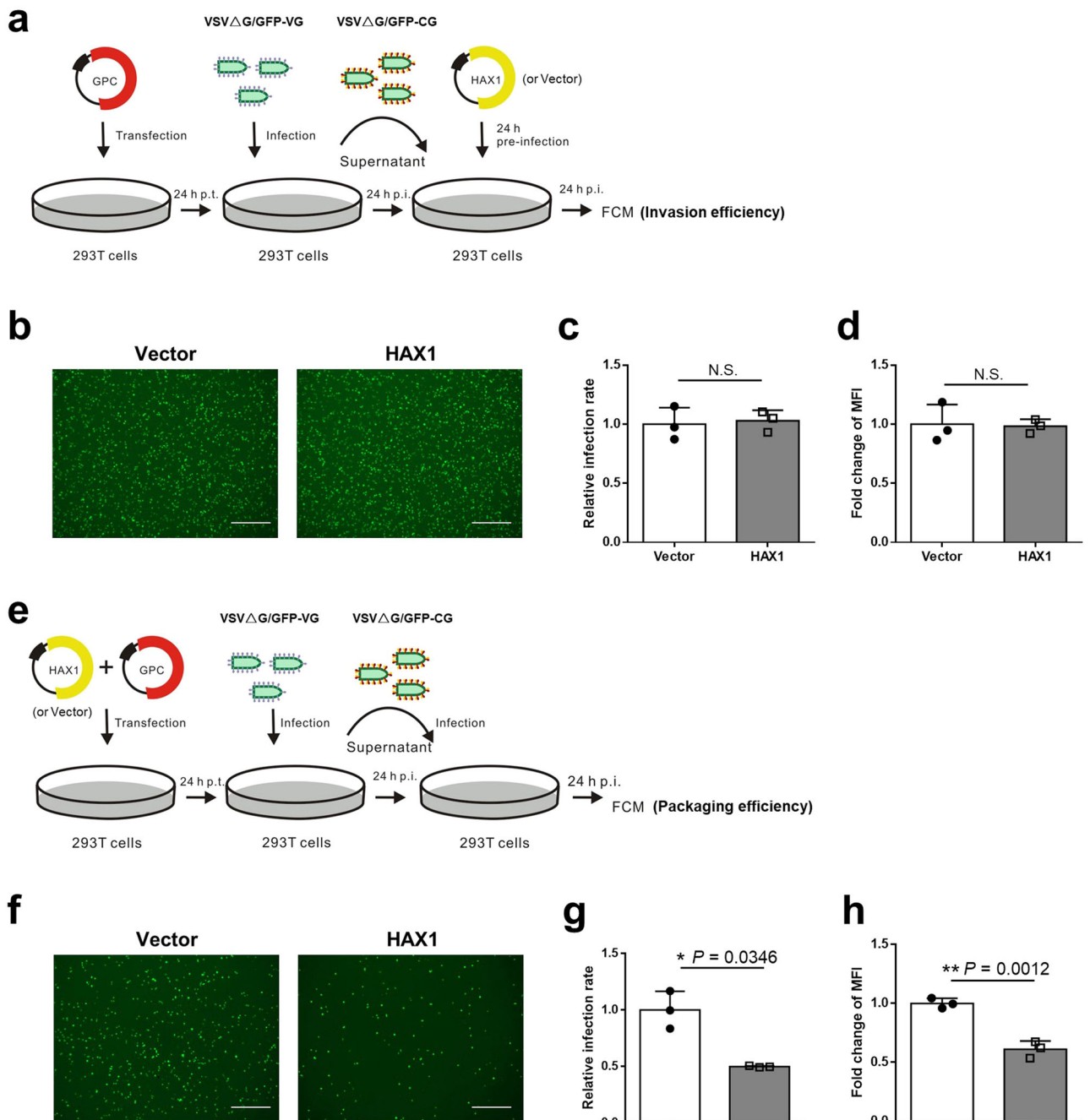

**Fig. 7 | HAX1 inhibits CCHFV glycoprotein-mediated packaging but not invasion. a**–**d** Effect of HAX1 on CCHFV glycoprotein-mediated invasion into cells. The strategy is illustrated in (**a**). CCHFV pseudotyped virus (VSV△G/GFP-CG) was generated as described in "Materials and methods" and then used to infect HEK293T cells pre-transfected with HAX1 expression plasmid or control empty vector. To determine the invasion efficiency of the pseudotyped virus, infected (GFP-positive) cells were visualized 24 h later by an inverse fluorescence microscopy (**b**) and quantified by flow cytometric analysis. Relative infection rate (**c**) and mean fluorescence intensity (MFI) (**d**) normalized to control are shown, respectively. **e**–**h** Effect of HAX1 on CCHFV glycoprotein-mediated packaging. The strategy is illustrated in (**e**). CCHFV pseudotyped virus (VSV△G/GFP-CG) was generated

in the contexts of co-transfection with the HAX1 expression plasmid or control vector and then used for infection of HEK293T cells as also described in "Materials and methods". Infected (GFP-positive) cells were visualized 24 h later by an inverse fluorescence microscopy (**f**) and quantified by flow cytometric analysis. Relative infection rate (**g**) and MFI (**h**) normalized to control are respectively shown. Bars, 50 μm. VSV△G/GFP-CG GFP-expressing pseudotyped VSV bearing CCHFV glycoprotein, VSV△G/GFP-VG GFP-expressing pseudotyped VSV bearing VSV G. Data are shown as means ± SD ($n = 3$ biologically independent samples). Comparisons were performed with unpaired two-tailed $t$-test (**c**, **d**, **h**) or $t$-test with Welch's correction (**g**). *$p < 0.05$; **$p < 0.01$; N.S. not significant. See also Supplementary Fig. S6. Source data are provided as a Source Data file.

HAX1 was previously also shown to interact with several other virus-encoded proteins and may play potential roles in virus-host interactions, although the mechanisms and significances need more investigations[35–39]. For example, HAX1 was shown to interact with the PA polymerase subunit of influenza A virus H1N1 and likely inhibit the

virus infection[39]. However, another group reported that HAX1 may play a beneficial role in influenza virus infection, as the authors found that the replication of avian influenza virus H9N2 was suppressed in HAX1-knockdown cells[40]. These studies suggest that HAX1 may be either negatively or positively regulate some viral infections, depending on

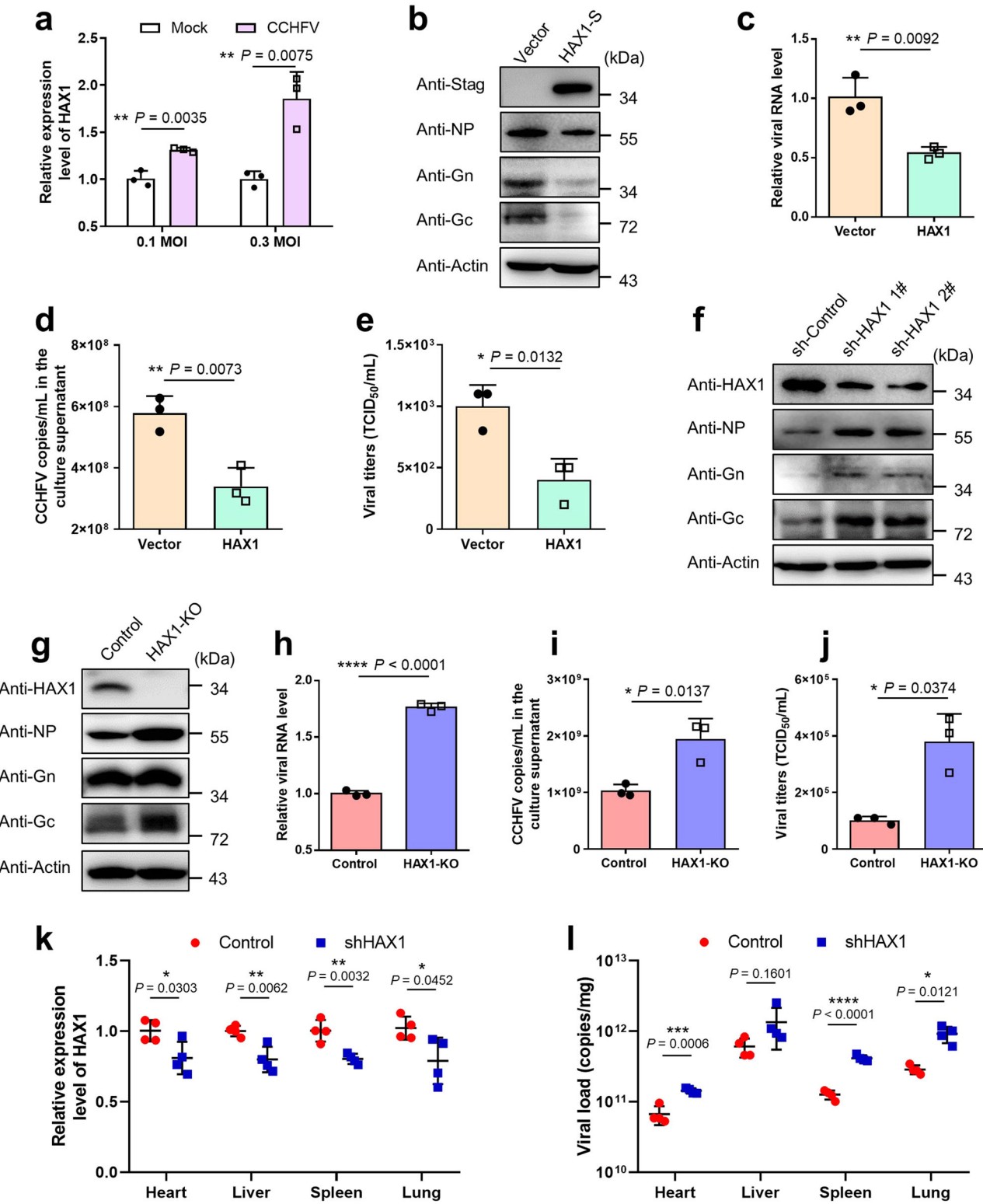

specific viruses, and more researches are required to determine its function and mechanism. Mitochondrion harbors multiple signaling proteins (especially MAVS) critical for innate immune responses and is known as a notable antiviral platform by contributing to the signal transduction of antiviral innate immunity. Here, with CCHFV as a model, we demonstrated an antiviral mechanism by HAX1 through hijacking of the viral envelope protein on mitochondria to abate virion packaging, expanding the knowledge of HAX1 biology and also perhaps defining a new paradigm of mitochondria as an antiviral platform

via HAX1. Bunyavirus is a large group of segmented negative-sense RNA viruses containing more than 500 members with CCHFV as a medically most important, high-pathogenic representative. It will be interesting to further test whether other bunyaviruses and their glycoproteins can be influenced by HAX1.

As aforementioned, research on CCHFV has long been restricted by the requirement of high-containment laboratories for experimental manipulation of living virus, which impedes the development of antiviral therapeutics and even many basic experimental materials and

**Fig. 8 | HAX1 acts as a host restriction factor against CCHFV infection. a** HAX1 expression is upregulated in response to CCHFV infection. HEK293 cells were infected with CCHFV at an MOI of 0.1 or 0.3 and then harvested at 24 h p.i. for evaluation of HAX1 mRNA levels. **b**–**e** HAX1 overexpression inhibits CCHFV infection. Huh7 cells transfected with HAX1-S expression plasmid or empty vector were infected with CCHFV at an MOI of 0.1. Viral proteins levels (**b**), relative replication levels of viral RNA segment (S) in infected cells (**c**), viral genome copies (**d**) and virus titers (**e**) in the culture supernatants were measured. **f** HAX1 KD by RNAi promotes CCHFV infection. Cells were transfected with the indicated shRNA-encoding or control plasmids and then infected with CCHFV at an MOI of 0.1. At 48 h p.i., indicated proteins levels were analyzed. **g**–**j** HAX1 KO by CRISPR-Cas9 promotes CCHFV infection. HAX1-KO or control HEK293 cells were infected with CCHFV at an MOI of 0.1 for 48 h and then detected for proteins levels (**g**), relative

viral RNA replication (**h**), viral genome loads (**i**), and viral titers (**j**). See also Supplementary Fig. S4. **k, l** 6–8 week-old female C57BL/6 mice ($n = 4$) were transduced with the viral vectors encoding HAX1-targeting or control shRNAs ($1 \times 10^7$ TU), followed by infection with CCHFV ($1.5 \times 10^6$ TCID$_{50}$). After 3 days post infection, HAX1 expression (**k**) and viral loads (**l**) in indicated tissues were quantified. In (**k, l**), values represent means ± SD ($n = 4$ mice). In (**a, c**–**e** and **h**–**j**), data are shown as means ± SD ($n = 3$ biologically independent samples). Comparisons were performed with unpaired two-tailed $t$-test in (**a, c**–**e, h, i** and **k**) or $t$-test with Welch's correction in (**j**). For comparisons in (**l**), unpaired two-tailed $t$-test ("heart" and "spleen") or $t$-test with Welch's correction ("liver" and "lung") were used, respectively. *$p < 0.05$; **$p < 0.01$; ***$p < 0.001$; ****$p < 0.0001$; N.S. not significant. Data are representative of three independent experiments with similar results (**b, f, g**). Source data are provided as a Source Data file.

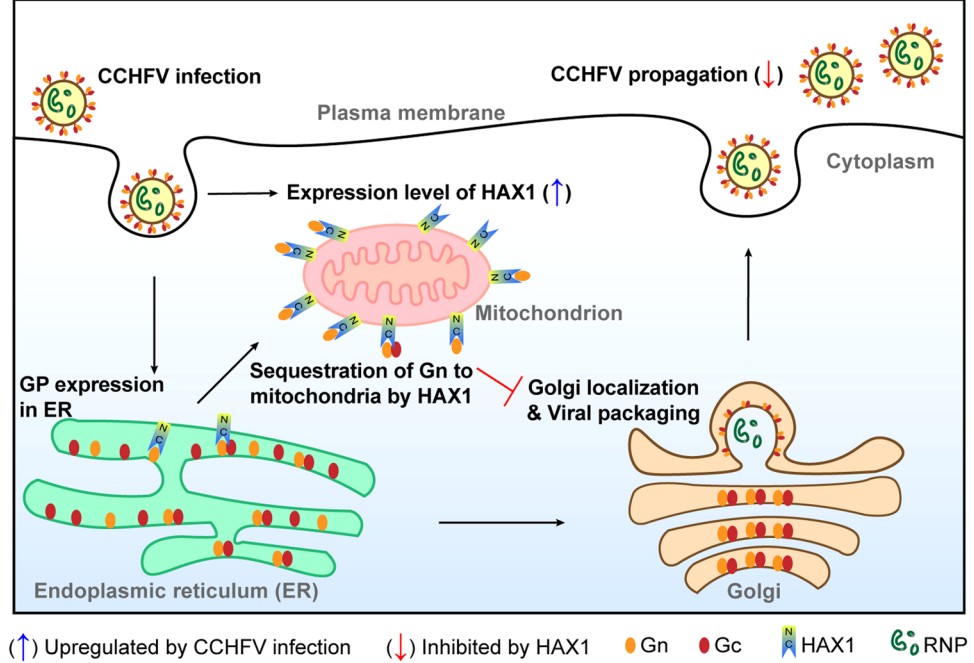

**Fig. 9 | Proposed model for host restriction of CCHFV via HAX1 by sequestrating Gn to mitochondria.** By its C-terminus interaction with Gn and N-terminus-mediated localization on mitochondria, HAX1 sequestrates Gn to mitochondria, thus disrupting the Golgi localization of Gn and inhibiting CCHFV packaging and propagation. Gc appears to be also hijacked by HAX1 to mitochondria (likely in the Gn/Gc complex form), but to a lesser extent. Additionally, HAX1 expression seems to be upregulated upon CCHFV infection, which in turn may act as an add-on for

host antiviral response via HAX1. Mitochondrion has been known as an innate immune antiviral platform harboring the mitochondrial antiviral signaling protein (MAVS) to function as the antiviral signaling hub. The findings here may recognize a new (HAX1-mediated) role of the mitochondrial platform in the host antiviral response apart from the MAVS-directed innate immunity. See also the text for more details.

platform such as infection models based on cultured cells and laboratory animals. In this study, we showed that knockout of HAX1 leads to significant increase of viral titers. Thus, HAX1-KO cell lines might be a useful tool for CCHFV amplification, which has always been the bottleneck of CCHFV research as it is difficult to amplify the virus with high titers using conventional cell lines[41]. Obtaining virus of high titers will advance many fundamental studies and inactivated vaccine development. Furthermore, knowledge of host restriction factors such as HAX1 may help inform development of new animal models, e.g., those by specific deletion of the restriction factors.

During the virus life cycle, the viral glycoproteins are supposed to drive virus entry, be newly produced and transported, and mediate progeny virion packaging along membranous organelles. Indeed, the interactome established here is well consistent with the supposed physiological actions of CCHFV glycoproteins (Fig. 3h). Thus, aside from HAX1, many other host proteins and their interactions with CCHFV glycoproteins identified here may also merit further studies.

The following are some additional implications of the established CCHFV glycoprotein interactomes.

Entry, the first step of virus infection, requires interactions between the viral envelope glycoproteins and the cellular attachment factors and receptors. Some plasma membrane proteins previously reported to be involved in cell entry of some viruses were identified here. For instance, BSG and HSPA5 have been identified as coreceptors for severe acute respiratory syndrome coronavirus 2 (SARS-CoV-2)[42,43]. HSPA5 also is required for the entry of several additional viruses such as dengue virus and Japanese encephalitis virus[44]. It will be interesting to investigate whether these interactors on the surface of host cell may contribute to cell invasion by CCHFV.

The ER is the site of CCHFV glycoprotein synthesis and processing, while the Golgi serves as the compartment for functional mature of the glycoproteins and critically, virion assembly by the glycoprotein-mediated packaging. Identification of those Gn/Gc-interacting host proteins located in the ER and Golgi apparatus, performing functions such as protein folding and trafficking, may provide

insights into the understanding of the complex biological processes of CCHFV glycoproteins. Mitochondria, as sites of the tricarboxylic acid (TCA) cycle and oxidative phosphorylation, are powerhouses for cellular energy production and also provide energy and macromolecular precursors for virus[45]. The interactions between CCHFV glycoproteins and mitochondrial proteins suggest that glycoproteins may be involved in these mitochondrial pathways. Interesting, many Gn/Gc interactors are located both in mitochondria and ER, indicating that communication between the two organelles may also play roles in viral infection.

In addition, many CCHFV Gn/Gc interacting host proteins are involved in ER or oxidative stress. Identification of oxidative stress-related proteins, such as the components of cytochrome c oxidase (COX5A, MT-CO2, and NDUFA4) which drive oxidative phosphorylation in the mitochondrial electron transport chain, may correspond to the upregulation of oxidative stress-response proteins induced by CCHFV infection[46]. The disorder of cellular redox equilibrium could also contribute to the pathogenesis of viral infections[46]. During virus infection, accumulation of viral proteins and competition with host proteins for modifications by viral glycoproteins may lead to ER stress. To alleviate this adverse effect, the cell operates an adaptive response known as the unfolded protein response (UPR) to reduce the load of newly synthesized proteins and eliminate inappropriately folded proteins through upregulation of ER chaperone expression. Besides, unassembled glycoproteins can be deployed to ubiquitylation and proteasomal degradation from ER by the ERAD pathway[47]. In our study, some Gn/Gc-interacting chaperone proteins are involved in ER stress and ERAD pathway, including members of HSP70 family (HSPA5, HSPA8, and HSPA1B), SSR4, THBS4, and UGGT1, which is consistent with ER stress caused by CCHFV infection as reported previously by us and others[48,49]. The data obtained here may provide clues and host targets for further depiction of the molecular mechanisms underlying CCHFV-infection-triggered stress responses.

Other notable host proteins interacting with CCHFV Gn/Gc include vesicle-associated membrane protein 2 (VAMP2) with functions in ER-Golgi trafficking, and importin subunit beta-1 (KPNB1), kinectin (KTN1) and UBB which drive the intracellular transport. Prohibitin (PHB), a mitochondrial chaperone which plays a critical role in several viruses, was detected in our interactome. PHB is implicated in neurovirulence of enterovirus 71 (EV71) and can serve as a potential drug target to limit neurological complications[50]. It is worth noting that neurological disorders are also documented in CCHF cases[51]. Interestingly, Urotensin 2 (UTS2), a potent vasoconstrictor involved in blood vessel diameter maintenance[52], has interaction with CCHFV Gn. The adhesive glycoprotein THBS4 and the transmembrane glycoprotein BSG identified in the Gn interactome may regulate vascular inflammation and angiogenesis as well[53,54]. Besides, ATP5A1, ATP5B and HSPB1 are known to positively regulate blood vessel endothelial cell migration. Further studies may be merited to explore whether these molecular functions might be related to endothelial damage and vascular leakage resulted by CCHFV infection. CCHFV is a typical hemorrhagic fever virus, causing clinical symptoms manifested by multiple organ hemorrhage. These findings might also help inform future studies on CCHFV pathogenesis and in particular, hemorrhagic mechanisms.

In summary, this study establishes the cellular interactomes of CCHFV glycoproteins and constructs a possible panorama of CCHFV glycoprotein-host interactions during the viral life cycle. CCHFV GPs may target to or be targeted by host factors to bolster or restrict viral infection, contributing to viral fitness or in turn host defense. Also, some of the interactions may interfere with normal cellular physiological activities to contribute to viral pathogenesis, meriting further investigations. Following the concept, we here characterize the interaction with HAX1 and unravel the interesting anti-CCHFV mechanism by HAX1. The present data provide plentiful clues for better

understanding of virus-host interactions and may help advance the development of antiviral therapeutic strategies.

## Methods

### Ethics statement
All animal experiment procedures were approved in advance by the ethics committees of Wuhan Institute of Virology, Chinese Academy of Sciences (approval number: WIVA33202207).

### Cell line and virus strain
Human embryonic kidney 293T cells (HEK293T, American Type Culture Collection [ATCC], CRL-3216) and human hepatocarcinoma cells (Huh7, National Virus Resource Center [NVRC], IVCAS 9.005) cells were grown in Dulbecco's modified Eagle's medium (DMEM) supplemented with 10% fetal bovine serum (FBS; Gibco, Grand Island, NY, USA). Human embryonic kidney 293 cells (HEK293, ATCC, CRL-1573) and human cervix adenocarcinoma cells (HeLa, ATCC, CCL-2) were cultured in Eagle's Minimum Essential Medium (EMEM) with 10% FBS. All cell lines grown at 37 °C in a humidified atmosphere of 5% $CO_2$. CCHFV strain YL16070 (GenBank accession no.: KY354082) was obtained from NVRC (IVCAS 6.6329) as previously described[55] and propagated in HEK293 cells. Experiments with living CCHFV infection were performed in the Biosafety Level 3 Laboratory, Wuhan Institute of Virology, Chinese Academy of Sciences. Viral titers were determined using 50% tissue culture infectious dose ($TCID_{50}$) method[41].

### Plasmids
For affinity purification, the coding sequence of CCHFV GPC from the prototype IbAr10200 strain (GenBank accession no.: NC_005300) was cloned into pCAGGS with a twin-Strep-tag on the C-terminus of Gc or on the N-terminus of Gn (after the SKI-1 cleavage site) to generate pCAGGS-Gn-Gc/2Strep or pCAGGS-2Strep/Gn-Gc, respectively (Fig. 1a). An expression plasmid encoding GPC with Gc tail truncated was used in the pseudotyped virus assays[22]. The coding sequence of human *HAX1* gene was amplified with cDNA from HEK293T cells after reverse-transcription polymerase chain reaction (RT-PCR). Then, full length or truncated HAX1 expression plasmids were generated by cloning the corresponding encoding sequences into pCAGGS with a C-terminal fused S tag. The coding sequence of human beta-1,4-galactosyltransferase 1 (*B4GALT1*) gene was amplified from cDNA of HEK293T cells and cloned into pEGFP-N1 to generate pEGFP-B4GALT1. All the expression plasmids were constructed by standard molecular biology techniques.

### Antibodies and reagents
Mouse monoclonal antibodies (mAb) against CCHFV Gc and mouse or rabbit polyclonal antibodies (pAb) against CCHFV Gn were prepared in house. Briefly, the nucleotide sequences encoding the ectodomains of Gn (residues 533–708 of GPC) and Gc (residues 1154–1598 of GPC) were cloned into pET28a vector in frame with His tag, respectively, and expressed in *Escherichia coli* BL21. The proteins were purified by affinity chromatography using nickel-charged resin and then used as antigens to generate polyclonal or monoclonal antibodies as described previously[56,57]. A mouse monoclonal antibody cell line (43E5) against CCHFV NP was generated in our lab as previously reported[58]. Antibodies against S-tag (Sino Biological, Cat#101290-T38, Dilution 1:2000), Strep-tag (Sangon Biotech, Cat#D191106, Dilution 1:2000), HAX1 (Proteintech, Cat#11266-1-AP, Dilution 1:1000), KPNB1 (ABclonal, Cat#A8610, Dilution 1:1000), RPN1 (Abcam, Cat#ab198508, Dilution 1:2000), HADHA (ABclonal, Cat#A13310, Dilution 1:1000), THBS4 (ABclonal, Cat#A16438, Dilution 1:1000), ATP1A1 (ABclonal, Cat#A7878, Dilution 1:1000), COXIV (Proteintech, Cat#11242-1-AP, Dilution 1:1000), GAPDH (Proteintech, Cat#10494-1-AP, Dilution 1:5000), and β-actin (Proteintech, Cat#66009-1-Ig, Dilution 1:5000) were purchased from the indicated manufacturers. Secondary

antibodies included horseradish peroxidase (HRP)-labeled goat anti-mouse (Proteintech, Cat#SA00001-1, Dilution 1:5000) or anti-rabbit IgG (Proteintech, Cat#SA00001-2, Dilution 1:5000) antibodies, Alexa Fluor-488 goat anti-mouse IgG (Abcam, Cat#ab 150113, Dilution 1:1000), Alexa Fluor-555 goat anti-mouse IgG (Abcam, Cat#ab 150114, Dilution 1:1000), Alexa Fluor-647 goat anti-rabbit IgG (Abcam, Cat#ab 150079, Dilution 1:1000), Hoechst 33258 (Beyotime, Cat#C1011), and MitoTracker red CMXRos (Thermo Fisher Scientific, Cat#M7512) were purchased from the indicated manufacturers, respectively.

## Cell viability assay

Cell viability was measured with Cell Counting Kit-8 assay (CCK8, MedChemExpress, Cat#HY-K0301). Briefly, cells transfected with CCHFV GPC expression plasmid or control vector were seeded in 96-well plates at a density of $1 \times 10^4$ cells/well. At 12, 24, and 36 h post transfection, 10 μl of CCK8 reagent (in 100 μl medium/well) was added and incubated with cells for 1 h at 37 °C. The absorbance at 450 nm was measured using a Biotek Synergy H1 Microplate Reader (Biotek). Survival rate of GPC overexpressing cells was normalized by comparison to the control.

## Strep-tag affinity purification and mass spectrometry analysis

For affinity-purification mass spectrometry (AP-MS), HEK293T cells were plated in 10-cm dishes and transfected with 15 μg (per dish) of pCAGGS-Gn-Gc/2Strep, pCAGGS-2Strep/Gn-Gc or the empty vector pCAGGS as a control using Lipofectamine 3000 (Invitrogen). At 36 h post-transfection (h p.t.), the cells were collected and lysed in 1 ml prechilled lysis buffer (20 mM Tris-HCl, 150 mM NaCl, 1 mM Ethylene Diamine Tetraacetic Acid [EDTA], 1% Triton X-100, and Complete protease inhibitor cocktail [Roche, Cat#04693132001], pH 7.5) for 30 min. After centrifugation at 4 °C for 15 min at 12,000 × g, the supernatants were incubated with 40 μl of MagStrep XT beads (IBA Life Sciences, Cat#2-4090-002) for 6 h at 4 °C. Then, the beads were washed with prechilled lysis buffer for three times, followed by washing with detergent-free wash buffer (20 mM Tris-HCl, 150 mM NaCl, and 1 mM EDTA, pH 7.5) for three times. The precipitated proteins were eluted from the beads by boiling in 1 × sodium dodecyl sulfate (SDS) loading buffer and then subjected to sodium dodecyl sulfate-polyacrylamide gel electrophoresis (SDS-PAGE), followed by silver staining or Western blot analysis.

After verifying protein expression and purification, the samples were subjected for label-free quantitative mass spectrometry[59,60]. Four independent biological replicates were performed for each Strep-tagged protein or the control in AP-MS analysis. Protein samples of Strep affinity purification were subjected to in-solution digestion with trypsin and obtained tryptic peptides were then analyzed by liquid chromatography-tandem mass spectrometry (LC-MS/MS) on a hybrid quadrupole-TOF mass spectrometer (TripleTOF 5600+, SCIEX) equipped with a nanoLC system (nanoLC-Ultra 1D Plus, Eksigent)[59,61]. Raw data from TripleTOF 5600+ were analyzed with MaxQuant (V1.6.2.10) Software[62] and searched against the UniProt human reference proteome database[63]. To identify CCHFV GPC peptides, the MS spectra were additionally searched against corresponding FASTA files for the viral proteins (Gn, Gc, mucin, GP38, and NSm). Search results were imported to Perseus software[64] for protein quantification and statistics analysis. The quantitative fold change (FC) of each protein compared to control and the significance (p value) of the t's exact test were calculated. Only proteins meeting the following two criteria were considered as Gn/Gc interacting host proteins: FC greater than 2 and p value less than 0.05.

## Bioinformatics analysis

To perform GO analysis, the ID list of host interactors was submitted to Protein Annotation Through Evolutionary Relationship (PANTHER) (http://www.pantherdb.org/) with total genes in the homo sapiens database set as the background. Proteins were classified into different categories based on their roles in cellular component, molecular function and biological process and a statistical over-representation test was performed using Fisher's exact test with False Discovery Rate (FDR) correction. The threshold value of FDR was set to 0.05. Redundant GO terms were then eliminated by REVIGO (http://revigo.irb.hr/) based on semantic similarity. The allowed similarity in REVIGO was set to "Medium (0.7)" and a semantic similarity measure of "SimRel" was selected. Proteomics data were analyzed and visualized with the statistical software environment R and R Studio. The identified host factors were submitted to STRING (https://string-db.org/) to qualify the protein-protein interactions, which were then visualized by Cytoscape (version number: 3.6.1).

## S-tag pulldown assay

S-tag pulldown assays (S-pulldown) were also used for protein interaction analysis[20,25]. Briefly, cells were transfected with the S-tagged protein expression constructs as indicated were harvested in the lysis buffer as described above. Then, the clarified cell lysates were incubated with S-protein agarose (Millipore, Cat#69704). After extensive washing, protein complexes were eluted from the beads by incubation in 1 × SDS loading buffer. Eluates and whole-cell lysates were analyzed by Western blot analysis with the antibodies as indicated.

## Western blot analysis

Protein samples from transfected or infected cells were subjected to SDS-PAGE and transferred to polyvinylidene difluoride (PVDF) membranes (Millipore, Cat#L3000015). After blocking with Tris-buffered saline (TBS) containing 5% nonfat milk, the membranes were incubated with primary antibodies at 4 °C overnight and subsequently with the corresponding HRP-conjugated secondary antibodies for 1 h at 37 °C. Protein signals were detected by SuperSignal West Pico Chemiluminescent Substrate (Thermo Scientific, Cat#34580).

## Immunofluorescence assay

Immunofluorescence assay (IFA) combined with confocal microscopy was performed to analyze protein subcellular localization. Briefly, transfected cells were fixed with 4% paraformaldehyde in phosphate-buffered saline (PBS) and incubated in 0.2% Triton X-100-PBS for permeabilization. After blocking with 5% bovine serum albumin (BSA, Biosharp) in PBS, cells were then treated with primary antibodies at 4 °C overnight, followed by incubation with the secondary antibodies for 1 h at room temperature. For visualization of the nuclei, the cells were incubated with Hoechst 33258 for 5 min at room temperature. To label mitochondria, the MitoTracker probe was added onto transfected cells at a final concentration of 100 nM prior to fixation. After 45 min incubation at 37 °C, the staining solution was removed, followed by IFA as described above. Images were gained and analyzed by an Andor Confocal Microscope (Dragonfly 202). For co-localization assay, Pearson's correlation coefficient (PCC) was calculated using coloc−2 plugin of the extended of ImageJ version Fiji (http://imagej.nih.gov/ij).

## Mitochondrial isolation

After transfected with indicated plasmids, -1.5 × 10^7 HEK293 cells were harvested and fractionated using a cell mitochondrial isolation kit (MedChemExpress, Cat#HY-K1060). In brief, the cells were suspended in mitochondrion isolation reagent (included in the kit) and placed in an ice bath for 15 min. Then, the cells were homogenized and centrifuged at 1000 × g for 10 min. The supernatant was collected and centrifuged again at 3500 × g for 10 min to precipitate the mitochondria. The isolated mitochondria were suspended in mitochondrion lysis buffer and subjected to Western Blot analysis.

## Analysis of CCHFV glycoprotein-mediated entry and packaging with pseudotyped viruses

To generate the CCHFV pseudotyped virus, i.e., VSV△G/GFP-CG (VSV bearing CCHFV glycoproteins and expressing GFP), HEK293T cells transfected with CCHFV GPC expression plasmid were infected at 24 h post-transfection with VSV△G/GFP-VG in which the *G* gene of VSV is replaced with *GFP*[22,23]. After 2 h of adsorption at 37 °C, the viruses were removed and cells were extensively washed three times with serum-free DMEM. After 24 h of incubation at 37 °C in DMEM supplemented with FBS, the culture supernatants were centrifuged and filtered through a 0.45-μm filter to remove cell debris, followed by storage at −80 °C until further experiment. To produce the control pseudotyped virus bearing VSV G, a VSV G expression plasmid was used for transfection. To determine the effect of HAX1 on the production of VSVΔG/GFP-CG or VSVΔG/GFP-VG, HAX1 expression plasmid pCAGGS-HAX1-S or empty plasmid pCAGGS were co-transfected with the CCHFV GPC or VSV G expression plasmids. Then, the obtained pseudoviruses were used to transduce cells. At 24 h post transduction, GFP-positive cells were analyzed with fluorescence microscopy and flow cytometry using a BD LSRFortessa flow cytometer (BD Biosciences) and FlowJo software (version 7.6).

To determine the effect of HAX1 on the entry process of VSV△G/GFP-CG or VSV△G/GFP-VG, cells were transfected with the HAX1 expression plasmid or empty vector. At 24 h post transfection, the cells were infected with the indicated pseudotyped viruses for 24 h, followed by analyses of GFP-positive cells with microscopy or flow cytometer as described above. A figure exemplifying the gating strategy is provided in Supplementary Fig. S7.

## Gene silencing

The HAX1 RNAi constructs were generated by cloning double-stranded oligonucleotides targeting specific sequences into the shRNA-expressing vector, pLKO.1[49,65]. The target sequences of each shRNA plasmid were listed in Supplementary Table S3. Huh7 cells were seeded into 24-well plates and transfected with shRNA plasmids targeting HAX1 or the control using Lipofectamine 3000 according to the manufacturer's protocol. At 36 h p.t., the cells were used for the following experiments and analyses.

## Generation of HAX1-KO cell lines by CRISPR-Cas9

The single-guide RNA (sgRNA) sequence targeting exo2 of HAX1 was designed with the online CRISPR Design tool (http://crispor.tefor.net/) and cloned into pSpCas9(BB)−2A-Puro (PX459) V2.0 vector[49,65]. HEK293 cells were transfected with the constructed pX459-based plasmids for 48 h and selected with 3 μg/ml puromycin (Gibco, Cat#A1113803) for 72 h. Then single clonal cell lines were isolated from the transfected cells by serial dilution, and identified by gene sequencing and Western blot analysis.

## CCHFV infection of HAX1-KD mice

Lentiviral particles used for HAX1-KD in mice were generated as previously described[25]. Briefly, HEK293T cells were transfected with the pLKO.1 shRNA plasmid and packaging plasmids psPAX2 and pMD2.G. At 48 and 72 h post-transfection, the cell culture media were collected by centrifugation at $1000 \times g$ for 5 min and filtered through a 0.45 μm filter to remove cell debris. Then, the supernatants were concentrated at $125,000 \times g$ (SW41 rotor; Beckman) for 90 min, and the pellets were resuspended in PBS and stored at −80 °C. To knock down the expression of HAX1 in vivo, 6–8-week-old female C57BL/6 mice ($n = 4$ per group) were injected with $1 \times 10^7$ transducing units (TU) of the HAX1-targeting or control viral vectors via caudal vein. One week after transduction, mice were challenged with CCHFV ($1.5 \times 10^6$ TCID$_{50}$) via intraperitoneal injections. At 3 days post infection, mice were humanely euthanized for tissue sample collection.

## RNA extraction and real-time qPCR assay

Infected cells or tissues were harvested and intracellular RNA was extracted using Trizol reagent (Invitrogen, Cat#15596018). Then RNA was subjected to reversed transcription using PrimeScript RT reagent Kit with gDNA Eraser (Takara, Cat#047A). Quantitative real-time PCR was performed with SYBR Premix Ex Taq II (Takara, Cat#820A) on StepOnePlus Real-Time PCR System (Applied Biosystems). Relative quantitation was performed using the $2^{−\Delta\Delta CT}$ method with β-actin or GAPDH as an internal control and the relative fold change was calculated by normalizing to control cells. Sequences of gene-specific primer pairs were listed in Supplementary Table S3.

Viral RNA from infected cell culture supernatant was extracted using the MiniBEST Viral RNA/DNA Extraction Kit (Takara, Cat#9766) according to the manufacturer's instructions. RNA was eluted in 30 μl RNase-free water. Real-time one-step qPCR was used to determine the viral loads with One Step PrimeScript T-PCR Kit (Takara, Cat#RR064A). The primers, probes and RNA standards were generated as previously described[66].

## Statistical analysis

Statistical analyses were performed by GraphPad Prism 8 (GraphPad Software, San Diego California USA, https://www.graphpad.com/). All results were expressed as mean ± SD. Statistical tests are individually specified within the figure legends. $p < 0.05$ was considered statistically significant. *$p < 0.05$; **$p < 0.01$; ***$p < 0.001$; ****$p < 0.0001$; N.S., not significant.

## Reporting summary

Further information on research design is available in the Nature Portfolio Reporting Summary linked to this article.

## Data availability

The raw MS proteomics data have been deposited to the Science Data Bank (CSTR: 31253.11.sciencedb.10567), and the access link is https://www.scidb.cn/en/s/Rn2INj. Other data are contained within the article/Supplementary Information. Source data are provided with this paper.

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

## Acknowledgements
This work was funded by the National Key Research and Development Program of China (2022YFC2303300, Y.J.N. and H.W.), the National Natural Science Foundation of China (32170171, Y.J.N.), the Youth Innovation Promotion Association of Chinese Academy of Sciences (Y.J.N.), and the Hubei Province Postdoctoral Funding Project (S.D.). We thank all team members of the National Virus Resource Center for the preserving viruses and cell lines used for experiment. We thank Mr. Jia Wu, Mr. Hao Tang, and Mr. Jun Liu from the running team of Biosafety Level 3 Laboratory, Wuhan Institute of Virology, Chinese Academy of Sciences for their critical support on experimental activities. We acknowledge Dr. Ding Gao, Ms. Anna Du, Ms. Juan Min, Ms. Pei Zhang, and Ms. Bichao Xu from the Institutional Center for Shared Technologies and Facilities of Wuhan Institute of Virology, Chinese Academy of Sciences and Dr. Yizi Liu, Dr. Qiangqiang Han, and Mr. Tong Liu from SpecAlly (Wuhan) for technical assistance.

## Author contributions
Y.N., F.D., and H.W. conceived the study. Y.N., S.D., and Y.M. designed the experiments. S.D. and Y.M. performed the majority of the experiments and acquired the data. Q.L., K.F., Z.J., Z.W., C.Z., J.Z., Y.F., Q.Z., F.R., and M.W. participated in experiments. S.D., Y.M., and Y.N. analyzed data and edited figures. Y.N. and S.D. wrote the original draft. Y.M. and Y.N. reviewed and edited the manuscript. Y.N., H.W., and S.D. provided funding. Y.N., F.D., and H.W. provided supervision. All authors reviewed and approved the manuscript.

## Competing interests
The authors declare no competing interests.
