## [Peer Review File · Nature Communications]

Interactome profiling of Crimean-Congo hemorrhagic fever virus glycoproteinsREVIEWER COMMENTS

Reviewer #1 (Remarks to the Author):

The manuscript uncovers proteins interacting with CCHFV Gn and Gc through proteomic analyses. It sheds light on potential cellular pathways that play a role in CCHFV assembly and pathogenesis, marking the first of its kind. The authors confirm some interacting protein hits through protein pulldown and western blotting with antibodies for identified host factors. Furthermore, they delve deeper into the mitochondrial protein interaction of HAX1 to showcase a specific interaction with Gn. This interaction leads to Gn's redistribution from the Golgi to the mitochondrion, which impedes CCHFV. The study reveals a modest reduction in viral yield in cell culture and mice.

Major points:

The manuscript's methods lack a clear definition of the constructs used for protein pulldown. To improve clarity, the exact location of the twin strep tags should be specified and included in both the manuscript methods and figure. Additionally, the schematic in Figure 1a would benefit from including Gn and Gc processing sites and Nsm if present. Currently, the figure indicates that non-structural glycoproteins (NSGs) were used as part of the bait protein for the pull. It is unclear whether GP38 or the mucin-like domain of NSGs were pulled down with either Gn or Gc baits. Please provide further details regarding the presence or absence of NSG peptide. Add NSG in figure 2 if present in the interactome.

In Figure 5, only data on Gn colocalization is provided. Since Gn and Gc interact, it would be valuable to perform the same experiment with a Gc antibody to determine how Gc is affected, if at all, by Gn redistribution to the mitochondrion. Additionally, the western blot presented in Figure 4 suggests that HAX1 affects PreGc maturation to Gc. However, the lane showing overexpression of HAX1 appears blurry and is difficult to analyze. It would be helpful if the authors could repeat the experiment and comment on whether PreGc or PreGn processing is affected by HAX-1. This is particularly important since GPC maturation is critical in CCHFV infectivity.

Figure 7 demonstrates that HAX1 affects packaging of CCHFV-VSV pseudotype. However, this difference was observed because the cell monolayer was not saturated with the virus. The control experiment in Supplementary Figure S1 shows that VSV-GP is not affected, as evidenced by the relative infection index. It is possible that the amount of pseudotype used in this experiment was much higher, resulting in the complete infection of the monolayer. To confirm this, dilutions of the VSV-G pseudotype should be performed to rule out any potential effects of saturation.

Minor points:

The strain of CCHFV used should be included in the methods.

It would be helpful to clarify in the text whether the VSV-pseudotype experiments used the complete GPC or a Gc tail deletion variant, and to which strain of CCHFV the GPC belongs.

Could you please provide more information or a reference on how the rabbit antibody against Gn was generated? It would be helpful to include detailed information about the immunogen used and the immunization protocol to ensure reproducibility and facilitate future studies.

Reviewer #2 (Remarks to the Author):

Dai and colleagues present in this manuscript several host interactors on CCHFV glycoproteins, and the potential antiviral effect of one of these. The paper is written in a clear style (except for a few long sentences, details below) and structured into two sections: first, the identification of host proteins potentially interacting with CCHFV glycoproteins Gn and Gc; second, in-depth analysis of HAX1, a mitochondrial protein that hijacks Gn, reducing its availability in the Golgi, hence hampering virus packaging and propagation.

The experiments are mostly well designed and presented. However, the conclusions arrived at are often overstated and confirmation-biased, especially in the interactome analysis section. The authors

should limit/adjust their conclusions to what the experimental conditions and the actual result indicates/suggests. Likewise, the section 'Global view of CCHFV GP-cell interactions' and large part of the discussion (i.e. from line 475) is highly speculative, forcing Gn/Gc in possible roles and involvement in cellular processes, with no substantial evidence to support this. The reach of these sections should be limited to the data presented here.

Major points of consideration:

1. Interaction of M segment-encoded accessory proteins with structural proteins are required for virus assembly. In particular, CCHFV Gn requires GP38 to escort and to allow Gc cleavage (Freitas et al., Plos Pathogen 2020, PMID: 32956404). Wouldn't be expected that GP38 (and possibly other NSGs) is one of the identified proteins after AP/MS? How do the authors explain that only cellular proteins and no viral proteins co-precipitate with Gc/Gn?

2. The authors assume that pulled-down proteins are only the result of direct interaction with Gn and/or Gc. However, host protein complexes can be pulled down by one of the interactors. This accounts for Gn and Gc themselves, which interact with each other and other NSGs. In addition, previous proteomic analysis revealed that HAX1 interacts with ATP synthase subunit beta (Kang et al., BBRC 2010, PMID: 20171186). The authors should take these elements into account to frame the scope of the interactome analysis. For instance, Figure 3e, 3f force Gn and Gc as central nodes for interactions and assumes that Gn/Gc interact independently with different subunits of the ATP synthase. In my opinion, these assumptions are not well supported by the experimental data. How could the authors discern direct to indirect interaction?

3. WB experiments of Gn and Gc (Figure 2d) show co-precipitation, expected due to the Gn-Gc interaction. This difference in the strep-pulldown (Figure 2d) for 2Strep/Gn-Gc is not reflected in Figure 2a, where both are shown with similar log₂FC values, indicating similar abundance. Could the authors comment on this discrepancy?

4. Section "Global view of CCHFV GP-cell interactions": this section is highly speculative. The physiological role of these possible interactions is not further identified and, as such, the authors include very ample cellular processes and several regulatory scenarios. Interaction of viral proteins with membrane, ER, mitochondrial, Golgi and cytoskeletal proteins is very likely. Similarly, the nature of these interactions can be either beneficial or detrimental for the virus. Identifying possible interactors is useful, but this section leads to no hypothesis, nor conclusions. Moreover, this is all assuming the results from the interactome where Gn and Gc are forced as central node, which, as discussed above, is inaccurate.

5. ER, cytoskeletal proteins and several subunits of the mitochondrial ATP synthase exhibit higher log₂FC values than HAX1 (Supplementary tables). Why did the authors decide to continue with HAX1?

6. Figure 4a (Line 247) demonstrates that Gn, superior to Gc, co-precipitates with HAX1. However, Figure 2a,b shows similar log₂FC values for HAX1, suggesting similar abundance in the pulldown, regardless the use of Gn or Gc as bait. How do the authors reconcile this contradiction?

7. Figure 4b is inconclusive and redundant with Figure 5a. Quantitative data of colocalization in Figure 4b is not provided. Conclusions on lower colocalization of Gc with HAX1 are therefore subjective, and not evident at all from the representative images.

8. Gn/Gc interactions with HAX1 should also be at place in GPC+vector transfected cells with endogenously expressed HAX1 (Figure 4b). In this case, Gn remains predominantly within the Golgi. Could the authors show colocalization with HAX1 to the mitochondria using commercially available antibodies for HAX1?

9. If HAX1 predominantly locates on mitochondria how would the hijacking occur from the pool of Gn on the Golgi during viral infection? The experiments from Figures 5 and 6 indicate that Gn colocalize to the mitochondria in the presence of HAX1. However, it is very likely that these

overexpression systems (driven by strong viral promoters) facilitate interaction of HAX1 and Gn in the cytosol or elsewhere, as these two proteins are produced in enormous quantities.

10. On the same note, mitochondrial staining with MitoTracker seems very diffused, covering the entire non-nuclear compartment. Like this, it's difficult to conclude that HAX1 is mitochondrial, and consequently, that Gn is hijacked to the mitochondria. Did the authors titrate the MitoTracker? Could the authors show that Gn is indeed enriched in the mitochondrial compartment after isolation of mitochondria by e.g. differential centrifugation?

Minor points:

1. The interactome analysis didn't reveal HAX1 as a host restriction factor. This is limited to identifying HAX1, among other proteins, as possible interactor with Gc and o Gn. I would recommend adapting the title accordingly.
2. The authors should include more information about the cellular functions of HAX1 and the potential implications for the cell in accumulating viral proteins in the mitochondria.
3. Line 157-160: as discussed above, pull down experiments are not conclusive for direct interactions between Gn or Gc, or the complex, with all identified proteins. The interaction could also be as part of a host complex. Moderate the conclusion here.
4. Lines 161, 187, 209: Sentences difficult to follow and understand. Please consider splitting.
5. Line 370: should be 'sacrificed'.
6. Line 409: there's no contradiction for 'However' in these lines.
7. Line 527: This is not a hypothesis, but a recount of all possible scenarios.

Reviewer #3 (Remarks to the Author):

Dai et al., describes an AP-MS based interaction proteomics approach to identify the host proteins interacting with the CCHFV glycoproteins Gn and Gc by expressing tagged viral proteins in HEK293 cells. The authors state that they aimed to find the functions of these glycoproteins - especially in terms of viral entry and assembly in addition to all the other cellular processes. The manuscript is well-written and the data seems to be of relevance.

I have a few comments that the authors could try to address:

1. Based on the literature, and as suggested by the authors, the choice of HEK293 makes sense. The authors have verified their primary findings using complementary methods to validate the findings. The main caveat is that given CCHFV can infect a number of cell lines (PMID: 33869079), the author's findings, though novel, is still limited to HEK293.
2. With reference to the presentation of the proteomics data, it would be useful to provide a ranked list of Gn/Gc interactors based on values such as iBAQ or spectral counts as a measure of the abundance. In experiments such as these using fold change may not be the best way to prioritize genes.
3. Did the over-expression of Gn/Gc affect the cell viability?
4. In Fig 2C, the gene names are not clearly visible. Please consider increasing the font size.
5. The authors state that the GO enrichment analysis of the Gn/Gc interactors showed multiple cellular processes and compartments. A majority were intracellular and Fig 3H shows some plasma membrane proteins as well that includes HSPA5. UniProt and other published data indicates HSPA5 is a cytoplasmic/ER protein. Can the authors clarify this?
6. One of the main aims of this study, as stated in the introduction was to identify proteins that facilitate viral entry and assembly. Did any of the plasma membrane proteins identified provide any hints in that direction?

POINT-BY-POINT RESPONSES TO REVIEWERS' COMMENTS

REVIEWER COMMENTS

Reviewer #1 (Remarks to the Author):

The manuscript uncovers proteins interacting with CCHFV Gn and Gc through proteomic analyses. It sheds light on potential cellular pathways that play a role in CCHFV assembly and pathogenesis, marking the first of its kind. The authors confirm some interacting protein hits through protein pulldown and western blotting with antibodies for identified host factors. Furthermore, they delve deeper into the mitochondrial protein interaction of HAX1 to showcase a specific interaction with Gn. This interaction leads to Gn's redistribution from the Golgi to the mitochondrion, which impedes CCHFV. The study reveals a modest reduction in viral yield in cell culture and mice.

Response: We sincerely thank the reviewer very much for all these valuable comments. By following the suggestions, we have conducted additional analyses and made revisions accordingly that further improve the paper quality and complement the study. Thanks a lot.

Major points:

The manuscript's methods lack a clear definition of the constructs used for protein pulldown. To improve clarity, the exact location of the twin strep tags should be specified and included in both the manuscript methods and figure. Additionally, the schematic in Figure 1a would benefit from including Gn and Gc processing sites and Nsm if present. Currently, the figure indicates that non-structural glycoproteins (NSGs) were used as part of the bait protein for the pull. It is unclear whether GP38 or the mucin-like domain of NSGs were pulled down with either Gn or Gc baits. Please provide further details regarding the presence or absence of NSG peptide. Add NSG in figure 2 if present in the interactome.

Response: We have further specified the exact location of the twin strep tags in the revised paper (lines 624-631, 1012-1022 and Figure 1a), as suggested. Also, we have labeled the cleavage sites of GPC for Gn, Gc and non-structural proteins (mucin, GP38 and NSm) in the schematic diagram in Figure 1a for clearer illustration. To identify the viral peptides, MS spectra were additionally searched against the corresponding FASTA files for the viral protein products, including Gn, Gc, mucin, GP38, and NSm. The nonstructural proteins encoded by CCHFV M genome, GP38 and mucin, (but not NSm), were also identified as Gn/Gc-interacting proteins, consistent with their potential assisted roles in the glycoprotein processing. Along with some complementary instructions (lines 165-168 and 679-681) in the revision, they have been added in the revised Figure 2a and 2b by following the suggestion. Many thanks.

In Figure 5, only data on Gn colocalization is provided. Since Gn and Gc interact, it would be valuable to perform the same experiment with a Gc antibody to determine how Gc is affected, if at all, by Gn redistribution to the mitochondrion. Additionally, the western blot presented in Figure 4 suggests that HAX1 affects PreGc maturation to Gc. However, the lane showing overexpression of HAX1 appears blurry and is difficult to analyze. It would be helpful if the authors could repeat the experiment and comment on whether PreGc or PreGn processing is affected by HAX-1. This is particularly important since GPC maturation is critical in CCHFV infectivity.

Response: By following the suggestion, we have performed the similar experiment of Gc and demonstrated that Gc is also hijacked by HAX1 to the mitochondrion, but to less extent compared with Gn (newly added Figure S2 in the revised paper). This is also consistent with the interaction of Gn and Gc. Additionally, the sequestration of Gn as well as Gc to mitochondria has been corroborated by mitochondrial fractionation (newly added Figure 5e). Indeed, it is possible that such an interactor might affect the processing of Gc, which, if so, could be an additional effect of HAX1. We really agree with that and have repeated the experiment per the suggestion. The conclusions of protein interactions are consistent; however, no notable influence of HAX1 on Gc or Gn production was observed, based on repeated experiments. A set of repeated data have been used in the revised version (Figure 4a) to replace the previous blots probably misleading by following the suggestion. Thanks a lot.

Figure 7 demonstrates that HAX1 affects packaging of CCHFV-VSV pseudotype. However, this difference was observed because the cell monolayer was not saturated with the virus. The control experiment in Supplementary Figure S1 shows that VSV-GP is not affected, as evidenced by the relative infection index. It is possible that the amount of pseudotype used in this experiment was much higher, resulting in the complete infection of the monolayer. To confirm this, dilutions of the VSV-G pseudotype should be performed to rule out any potential effects of saturation.

Response: Thanks for the comment. The packaging efficiency by VSV-GP is itself indeed higher than that by CCHFV GP. However, the infection ratios of the cell monolayers by even VSV-GP pseudotyped viruses in Figure S1 were actually around 15% ~ 45% in multiple repeated experiments. Thus, there was no complete, saturated infection of the monolayer in these assays. By following the suggestion, in a set of additional repeated experiments, we have tested the infection with diluted VSV-GP pseudotyped viruses as well and obtained consistent results that the control VSV-GP was not affected in the revised version (new Figure S6), also confirming the conclusion. Many thanks.

Minor points:

The strain of CCHFV used should be included in the methods.

Response: It has been included in the methods (lines 602-603 and 609-610). Thanks.

It would be helpful to clarify in the text whether the VSV-pseudotype experiments used the complete GPC or a Gc tail deletion variant, and to which strain of CCHFV the GPC belongs.

Response: Thanks for the kind reminder. The GPC of the prototype CCHFV IbAr10200 strain (GenBank accession no.: NC_005300) was used in the present study and a Gc tail deletion variant was used in the VSV-pseudotype experiments according to the previous reference. These have been clarified in the Methods of the revised paper (lines 609-614) by following the comment. We also tried to clone GPC of other strains which however, appeared difficult to be steadily constructed to meet the experimental requirements. Many thanks.

Could you please provide more information or a reference on how the rabbit antibody against Gn was generated? It would be helpful to include detailed information about the immunogen used and the immunization protocol to ensure reproducibility and facilitate future studies.

Response: Thanks for the suggestion. More detailed information of the antibodies, together with the related references, have been provided in the Methods (lines 624-631) according to the suggestion. Again, we really thank the reviewer very much for all these important suggestions that have helped us further improve and complement the paper.

Reviewer #2 (Remarks to the Author):

Dai and colleagues present in this manuscript several host interactors on CCHFV glycoproteins, and the potential antiviral effect of one of these. The paper is written in a clear style (except for a few long sentences, details below) and structured into two sections: first, the identification of host proteins potentially interacting with CCHFV glycoproteins Gn and Gc; second, in-depth analysis of HAX1, a mitochondrial protein that hijacks Gn, reducing its availability in the Golgi, hence hampering virus packaging and propagation.

The experiments are mostly well designed and presented. However, the conclusions arrived at are often overstated and confirmation-biased, especially in the interactome analysis section. The authors should limit/adjust their conclusions to what the experimental conditions and the actual result indicates/suggests. Likewise, the section 'Global view of CCHFV GP-cell interactions' and large part of the discussion (i.e. from line 475) is highly speculative, forcing Gn/Gc in possible roles and involvement in cellular processes, with no substantial evidence to support this. The reach of these sections should be limited to the data presented here.

Response: We are very grateful for all these valuable comments. By following the suggestions, we have made some revisions to the writing especially by adjusting or modifying some descriptions of the biological implications based on bioinformatics analyses and reference review. Mainly, previous contents of the section 'Global view of CCHFV GP-cell interactions' have been largely deleted and adjusted in the revised paper (lines 208-260, see also below). Also, a series of additional experimental analyses have been conducted per the suggestions, presenting consistent and strengthened results. We consider that the revision based on the suggestions have further complemented the study and improved the paper quality, making it more suitable for publication. Thanks a lot for your kind help.

Major points of consideration:

1. Interaction of M segment-encoded accessory proteins with structural proteins are required for virus assembly. In particular, CCHFV Gn requires GP38 to escort and to allow Gc cleavage (Freitas et al., Plos Pathogen 2020, PMID: 32956404). Wouldn't be expected that GP38 (and possibly other NSGs) is one of the identified proteins after AP/MS? How do the authors explain that only cellular proteins and no viral proteins co-precipitate with Gc/Gn?

Response: Thanks for the kind reminder. When we analyzed the MS results, the MS spectra were also searched against the nonstructural proteins and yes, GP38 and mucin (but not NSm) were identified as Gn/Gc-interacting proteins as well. This is indeed consistent with the previous study. The proteins and the reference by Freitas et al. that were inadvertently missed out have been included in Figure 2a and 2b and the Reference list (ref 15) of the revised manuscript, respectively. Many thanks.

2. *The authors assume that pulled-down proteins are only the result of direct interaction with Gn and/or Gc. However, host protein complexes can be pulled down by one of the interactors. This accounts for Gn and Gc themselves, which interact with each other and other NSGs. In addition, previous proteomic analysis revealed that HAX1 interacts with ATP synthase subunit beta (Kang et al., BBRC 2010, PMID: 20171186). The authors should take these elements into account to frame the scope of the interactome analysis. For instance, Figure 3e, 3f force Gn and Gc as central nodes for interactions and assumes that Gn/Gc interact independently with different subunits of the ATP synthase. In my opinion, these assumptions are not well supported by the experimental data. How could the authors discern direct to indirect interaction?*

Response: Sincere thanks for the comment. We really agree that the interactions identified could be direct or indirect by protein complexes. Moreover, protein complex-mediated indirect interactions could be also biologically significant and merit further investigations. These proteomics methods with high resolution and throughput recognize the interactors of high confidence but do not distinguish between direct or indirect interactions. Thus, we cannot assume that the pulled-down proteins are only the result of direct interaction. According to this and the following kind suggestions, we have modified Figure 3e and 3f by removing Gn and Gc to avoid potential misrepresentation of Gn and Gc as the cores and also added some supplementary words for more precise description (lines 166-168). Many thanks.

3. WB experiments of Gn and Gc (Figure 2d) show co-precipitation, expected due to the Gn-Gc interaction. This difference in the strep-pulldown (Figure 2d) for 2Strep/Gn-Gc is not reflected in Figure 2a, where both are shown with similar log₂FC values, indicating similar abundance. Could the authors comment on this discrepancy?

Response: Thanks for the comment. In Figure 2a, the fold change (FC) values of Gn and Gc were actually 311.1141 (i.e., log₂FC = 8.2813) and 204.7188 (i.e., log₂FC = 7.6775), respectively, when Gn was used as the bait protein. Thus, the signal of Gn here was stronger than that of Gc. Additionally, identification by mass spectrometry is based on the quantification of peptides; however, unlike host proteins, viral proteins are only present in samples transfected with GPC but not in the control samples. It might lead to a difference of the quantification for the viral proteins due to the much larger size and more lytic peptides of Gc, compared with Gn, and some potential preference of Gn/Gc peptide detection in mass spectrometry despite its high resolution. Besides, the coprecipitation of Gc with Gn seems sort of stronger than that of Gn with Gc as seen in Figure 2d, which may suggest a reciprocal (but possibly not completely equal) interaction and interestingly is consistent with the mass spectrometry quantification in Figure 2a and 2b.

4. *Section “Global view of CCHFV GP-cell interactions”: this section is highly speculative. The physiological role of these possible interactions is not further identified and, as such, the authors include very ample cellular processes and several regulatory scenarios. Interaction of viral proteins with membrane, ER, mitochondrial,*

Golgi and cytoskeletal proteins is very likely. Similarly, the nature of these interactions can be either beneficial or detrimental for the virus. Identifying possible interactors is useful, but this section leads to no hypothesis, nor conclusions. Moreover, this is all assuming the results from the interactome where Gn and Gc are forced as central node, which, as discussed above, is inaccurate.

Response: Thanks for the suggestion. Indeed, proteomics profiling may help outline the landscape of virus-host interactions; however, more specific investigations which might be guided by the proteomics data are needed for further validations in the future. By following the suggestion, we have removed the section and only retained a brief description of mapping the host proteins to virus life cycle concerning the supposed Gn/Gc actions (lines 208-260, 1069 and Figure 3h in of the revised paper), through GO (especially biological process and cellular component) analyses, together with literature review. We consider that this processing of the proteomics data might be helpful for understanding of the set data of various host proteins. Also, per the suggestion, we have revised Figure 3e and 3f as aforementioned to avoid potential misunderstanding of Gn and Gc as the central nodes. Many thanks for the guidance.

5. ER, cytoskeletal proteins and several subunits of the mitochondrial ATP synthase exhibit higher log₂FC values than HAX1 (Supplementary tables). Why did the authors decide to continue with HAX1?

Response: Thanks for the comment. As we discussed and you kindly mentioned above, based on the interactomes established here, Gn/Gc interactors are well linked to membrane (including plasma membrane), ER, mitochondrion, Golgi, vesicles, and cytoskeleton, which is consistent with the supposed Gn/Gc physiological actions and also hents some unexpected specific interactions such as that with HAX1. The following independent interaction validations of representative host factors, along with the function/mechanism studies on HAX1, further support the proteomics profiling. These suggest that the proteomics data reported here likely provide valuable clues for further resolution of the functioning and interactions of Gn/Gc (such important viral proteins) with host cells and could be of interest to the field for the future study. We considered HAX1 as an interesting and representative one and subsequently presented a new mitochondrion/HAX1-associated antiviral mechanism targeting the viral glycoproteins (especially Gn, the driver of packaging) and the mediated virion production. Indeed, we are also investigating other interactors discovered here and have found interesting functions which however, are still being validated and worked on. Thanks a lot.

6. Figure 4a (Line 247) demonstrates that Gn, superior to Gc, co-precipitates with HAX1. However, Figure 2a,b shows similar log₂FC values for HAX1, suggesting similar abundance in the pulldown, regardless the use of Gn or Gc as bait. How do the authors reconcile this contradiction?

Response: Thanks for the comment. The FC values of HAX1 were ~7.6 (i.e., log₂FC = 2.9) or 12.2 (log₂FC = 3.6) when Gc (Figure 2b and Table S2) or Gn (Figure 2a and

Table S1) was used as the bait proteins, respectively, indicating a stronger interaction of HAX1 with Gn. It is basically consistent with the independent interaction and WB analyses in the following Figure 2d and newly added Figure 2e that also show the stronger interaction with Gn. Figure 4a actually presents a reciprocal interaction analysis with HAX1 as the bait. Thus, these results demonstrate consistently that HAX1 interaction with Gn is stronger. According to the comment, we have also added a supplementary description (lines 273-275) in the revised version. In agreement with these, the following analyses including newly included experimental data based on the suggestions further show a likely stronger colocalization of HAX1 with Gn and more notable changes of Gn subcellular localization by HAX1.

7. Figure 4b is inconclusive and redundant with Figure 5a. Quantitative data of colocalization in Figure 4b is not provided. Conclusions on lower colocalization of Gc with HAX1 are therefore subjective, and not evident at all from the representative images.

Response: Thanks for the suggestion. The data in Figure 5 are further and progressive analyses with a new, advanced focus based on the clues in Figure 4. By following the suggestions, we have quantified the colocalization in Figure 4b and provided the data in a newly added Figure 4c in the revised version. These results are basically consistent with the mass spectrometry quantification and multiple interaction validations (Figure 2a, 2b, 2d, 2e, and 4a) and the following analyses including those newly added per the suggestions that show notable influence on localization of the viral proteins (especially Gn). Many thanks.

8. Gn/Gc interactions with HAX1 should also be at place in GPC+vector transfected cells with endogenously expressed HAX1 (Figure 4b). In this case, Gn remains predominantly within the Golgi. Could the authors show colocalization with HAX1 to the mitochondria using commercially available antibodies for HAX1?

Response: Thanks for the suggestion. Interactions of Gn/Gc with endogenously expressed HAX1 could be found in the mass spectrometry and independent interaction validations (Figure 2d and 2e). Also, by following the suggestion, we have added new data that show the partial but visible colocalization of Gn (as well as Gc) with endogenous HAX1 to the mitochondria (positive PCC values) (Figure S3), albeit to less extents compared to that in the context of overexpressed HAX1. More importantly, according to the suggestions below, we also have conducted mitochondrial fractionation to analyze the effect of HAX1 on Gn and Gc localization, which further evidently confirms the hijacking of the viral proteins (especially Gn) by HAX1 (including endogenously expressed HAX1) to mitochondria (please see the following responses and newly added Figure 5e and S5). Thanks a lot.

9. If HAX1 predominantly locates on mitochondria how would the hijacking occur from the pool of Gn on the Golgi during viral infection? The experiments from Figures 5 and 6 indicate that Gn colocalize to the mitochondria in the presence of HAX1. However, it is very likely that these overexpression systems (driven by strong

viral promoters) facilitate interaction of HAX1 and Gn in the cytosol or elsewhere, as these two proteins are produced in enormous quantities.

Response: Thanks for the comment. Although Gn is able to locate to the Golgi, exhibiting a substantial location at Golgi essential for its physiological function when detected by IFA, the viral protein is initially synthesized in ER, before the transport to the Golgi by the ER-Golgi trafficking pathway via vesicles. Interestingly, despite the main mitochondrial localization, HAX1 can be also detected in ER and cellular vesicles, albeit to less extents, and there are studies reporting its potential of vesicular trafficking (Suzuki et al., 1997, J Immunol, PMID: 9058808; Zhang et al., 2021, Nat Commun, PMID: 33741962; Pisani et al., 2021, Exp Cell Res, PMID: 33417922; Ortiz et al., 2004, J Biol Chem, PMID: 15159385; etc.). However, its predominant mitochondrial localization suggests the dominantly stronger localization and final retention to the mitochondria. It could be inaccurate to state the hijacking from the Golgi in the previous manuscript and the localization changes from Golgi to mitochondria is likely a result we observed but not the sequential process. The viral protein is more likely bound and hijacked by HAX1 from the cellular sites (especially ER) where they both exist. We have added more instruction and discussions and made minor revisions regarding this point (lines 448-453 and 460-468) according to the suggestions. Based on extensive observations during our study, the proteins exhibited clear and proper subcellular localization that are consistent with the previous reports even upon high expression levels. Some overexpressed proteins sometimes may be incorrectly folded and form random aggregates. There was no such hint possibly resulted by overexpression. Additionally, as discussed above, aside from the results of the viral protein interactions with endogenous HAX1 (Figure 2a, 2b, 2d and newly added 2e, and Table S1 and S2), data of partial localization on mitochondria in the context of endogenously expressed HAX1 have been further shown (newly added Figure S3). More importantly, by following the suggestions (including the next comment), we have also conducted mitochondrion isolation assay. It is complementary and indeed, as an analysis of cell population, has advantage of signal enrichment compared to IFA and confocal analyses, given the increased difficulty to note localization changes in relatively small/partial extents by observing individual cells and the requirements of high-quality antibodies and experimental settings when using IFA and confocal microscopy. By the suggested methodology, we have further shown not only the evident mitochondrial localization/enrichment of Gn, and likely to a lesser extent Gc, in the context of endogenous HAX1, but also the increase by higher HAX1 expression and decrease (abolishment of the enrichment) to mitochondria by HAX1 knockout in the revision (newly added Figure 5e and S5). Together, these results are consistent and further substantiate HAX1 hijacking of the viral glycoproteins (especially Gn) to mitochondria, strengthening and complementing the paper. Thank you very much.

10. On the same note, mitochondrial staining with MitoTracker seems very diffused, covering the entire non-nuclear compartment. Like this, it's difficult to conclude that HAX1 is mitochondrial, and consequently, that Gn is hijacked to the mitochondria. Did the authors titrate the MitoTracker? Could the authors show that Gn is indeed enriched in the mitochondrial compartment after isolation of mitochondria by e.g. differential centrifugation?

Response: We are sorry for the images with reduced resolution in the PDF file we previously provided. The original staining images were actually clear but greatly reduced due to the combination into Figures and the previous conversion to PDF. In resubmission, high-resolution Figures have been individually provided. The MitoTracker, a widely acknowledged, high-quality reagent for mitochondrial staining, is generally used in literature and in our lab, (an early usage example in our lab, Ning et al., 2014, PMID: 24706939). Based on the long-time usage, we usually perform the staining and obtain good results with the recommended dosage and conditions in the manufacturer's instruction, although other settings were also tried by us. A set of representative original images (derived from Figure 5b) showing the clear mitochondrial staining (mitochondrial dots and networks) and predominant mitochondrial localization of HAX1 together with Gn could be seen in the following Figure R1. Per the comment, here we have also conducted a comparative staining using the MitoTracker or IFA with anti-COXIV antibody and the corresponding labeled secondary antibody. As seen in the following Figure R2, the two staining methods exhibit clear and consistent mitochondrial morphology. Additionally, as responded above, mitochondrial fractionation assays have been conducted by following the suggestion and show consistent and complementary results, further substantiating and improving the study. Thank you very much.

Figure R1. Representative staining of mitochondria and the indicated proteins. These enlarged images were derived from Fig 5b (lower panel) for review.

Figure R2. Comparison of mitochondrial staining by MitoTracker or by IFA with antibodies. Cells were stained with MitoTracker red CMXRos prior to fixation, followed by detection of COXIV by IFA and confocal microscopy.

Minor points:

1. *The interactome analysis didn't reveal HAX1 as a host restriction factor. This is limited to identifying HAX1, among other proteins, as possible interactor with Gc and o Gn. I would recommend adapting the title accordingly.*

Response: Thanks for this kind suggestion. We have accordingly changed the title to “Interactome profiling of Crimean-Congo hemorrhagic fever virus glycoproteins and identification of HAX1 as a host restriction factor against the virion packaging” in the revised version.

2. *The authors should include more information about the cellular functions of HAX1 and the potential implications for the cell in accumulating viral proteins in the mitochondria.*

Response: Thanks. By following the suggestion, more discussions together with additional references regarding this point have been incorporated in the revision (lines 448-453 and 460-468).

3. *Line 157-160: as discussed above, pull down experiments are not conclusive for direct interactions between Gn or Gc, or the complex, with all identified proteins. The interaction could also be as part of a host complex. Moderate the conclusion here.*

Response: Thanks for the comment. By following the suggestion, an additional statement has been included in the revised paper to instruct that the interactions could be direct or indirect by complexes (lines 165-168), although both types of interactions could have biologically significant consequences on the viral proteins. Additionally, as aforementioned, minor revisions have also been made to the interaction network by removing the baits themselves in the revised paper (revised Figure 3e and 3f) to solely show the interactions documented in the STRING database per the suggestion.

4. *Lines 161, 187, 209: Sentences difficult to follow and understand. Please consider splitting.*

Response: We really appreciate this kind suggestion. Corresponding revisions have been made to these sentences (lines 172-175 and 200-204) and the last one has been deleted, by following the suggestions.

5. *Line 370: should be 'sacrificed'.*

Response: It has been corrected (line 412). Thanks for the kind reminder.

6. *Line 409: there's no contradiction for 'However' in these lines.*

Response: "However" has been deleted accordingly. Thanks.

7. *Line 527: This is not a hypothesis, but a recount of all possible scenarios.*

Response: It has been revised based on the comment (line 579). We would like to express our sincere thanks again for all these valuable suggestions which have helped us further significantly complement the study and improve the paper quality.

Reviewer #3 (Remarks to the Author):

Dai et al., describes an AP-MS based interaction proteomics approach to identify the host proteins interacting with the CCHFV glycoproteins Gn and Gc by expressing tagged viral proteins in HEK293 cells. The authors state that they aimed to find the functions of these glycoproteins - especially in terms of viral entry and assembly in addition to all the other cellular processes. The manuscript is well-written and the data seems to be of relevance.

Response: We really thank the reviewer very much for all these important comments and suggestions. By following them, we have further improved the paper in the revision. Many thanks.

I have a few comments that the authors could try to address:

1. Based on the literature, and as suggested by the authors, the choice of HEK293 makes sense. The authors have verified their primary findings using complementary methods to validate the findings. The main caveat is that given CCHFV can infect a number of cell lines (PMID: 33869079), the author's findings, though novel, is still limited to HEK293.

Response: We agree that considering multiple factors, HEK293/HEK293T can be considered as the top-priority cell type for the first-time establishment of the viral protein interactomes, while by using the interaction proteomics approach established here, analyses in other cells may be interesting and merited in the future as well. In the revision, although we did not generate new datasets using other cells, we have tested and validated the representative interactions in another related cell type, hepatic Huh7 cells, by following the suggestions (newly included Figure 2e of the revised version). As shown in our previous study mentioned by the reviewer (PMID: 33869079), Huh7 is another permissive cell to CCHFV infection (despite its lower transfection and bait expression efficiency compared to HEK293/HEK293T) and liver is also significantly related to CCHF. As presented in the newly added Figure 2e, the representative interactions identified and verified in HEK293T were well validated in Huh7 cells as well, further supporting and enriching the analyses. Thanks a lot.

2. With reference to the presentation of the proteomics data, it would be useful to provide a ranked list of Gn/Gc interactors based on values such as iBAQ or spectral counts as a measure of the abundance. In experiments such as these using fold change may not be the best way to prioritize genes.

Response: Sincere thanks for this detailed comment. The fold change is actually MaxLFQ fold change (ref 61, PMID: 24942700) quantified based on label-free quantification (LFQ) intensity. It is performed for inter-sample correction and used for inter-sample comparison. In comparison, iBAQ is based on the original intensity and mainly used for approximate comparison of different proteins within a same sample. For instance, in an experimental group, a protein ranked on the top by iBAQ may be

also existing and ranked high in the control group. Then it may not be recognized as a specific interactor unless it has a relatively higher abundance in the experimental group (by inter-group comparison to the control group). This study was conducted to identify the specific interactors of Gn/Gc by comparison to the control group (3 groups including Gn, Gc, and the control pulldown products, $n = 4$ for each group, and 12 samples in total). Thus, LFQ fold change was used for inter-group comparison (over the control group). Compared to spectral counting, the intensity-based LFQ measure avoids stochastic effects in ion sampling and performs inter-sample corrections to reduce the influences of possible differences in treatment, loading, presort, or instruments. It is likely more accurate and potentially provides higher reproducibility, despite the requirement of specific algorithms and software/hardware which however, is not a problem in recent years. We thus chose the MaxQuant LFQ method. Many thanks.

3. Did the over-expression of Gn/Gc affect the cell viability?

Response: We did not observe any noticeable effect of CCHFV GPC expression on the cell viability in our study. According to the comment, we have performed a CCK8 assay to verify the observation in the revision. Consistently, there was no significant difference in the cell viability between the groups with CCHFV GPC expression and the control (newly added Supplementary Figure S1). Thanks a lot for the suggestion.

4. In Fig 2C, the gene names are not clearly visible. Please consider increasing the font size.

Response: Thanks for the kind reminder. It has been addressed accordingly.

5. The authors state that the GO enrichment analysis of the Gn/Gc interactors showed multiple cellular processes and compartments. A majority were intracellular and Fig 3H shows some plasma membrane proteins as well that includes HSPA5. UniProt and other published data indicates HSPA5 is a cytoplasmic/ER protein. Can the authors clarify this?

Response: Thanks for the comment. Protein localization is sometimes variable by cell type and state, and Fig 3h shows several major subcellular locations of the identified host proteins likely involved in the Gn/Gc actions and virus life cycle based on the GO analysis and multiple databases such as UniProt, Swiss-Prot, and Genecards. Indeed, HSPA5 is primarily referred to as an ER molecular chaperone participating in processing and quality control of some proteins. However, according to gene ontology, the databases and literature, HSPA5 might be found in multiple compartments including ER (GO:0005783), cytoplasm (GO:0005737), mitochondria (GO:0005739), plasma membrane and cell surface (GO:0009986 and 0005886), Golgi (GO:0005793), etc. Interestingly, during cellular responses, HSPA5 could translocate from ER to plasma membrane, where it activates multiple signal transduction pathways (PMID: 32841659, 33960608, 33129921). Moreover, cell-surface HSPA5 has been reported as the port of entry for several viruses, such as dengue virus, enterovirus 71,

and SARS-CoV-2 (PMID: 33960608). Thus, it will be very interesting to further investigate the potential roles of these Gn/Gc-interacting host factors including HSPA5 in CCHFV infection, which also remain unexplored. A related discussion might be seen in the manuscript (lines 521-524 in the revised version).

6. One of the main aims of this study, as stated in the introduction was to identify proteins that facilitate viral entry and assembly. Did any of the plasma membrane proteins identified provide any hints in that direction?

Response: Thanks for this comment. Despite such pivotal roles of CCHFV Gn/Gc, their interactions with host cells long remain largely unknown and urgently need to be addressed. The present study for the first time analyzed systematically the interactions in proteomics perspective and identified many Gn/Gc interactors that may regulate or be regulated by Gn/Gc. These established interactomes thus may provide clues for further addressing long-unanswered questions of great importance concerning not only the host factors facilitating viral entry or assembly but also those inhibiting these processes, regulating other aspects of the glycoprotein actions, or being regulated by the glycoproteins, to contribute to the viral infection, pathogenesis, fitness or host antiviral response. Here we further elucidated the effect of a representative interactor HAX1 and a new HAX1/mitochondrion-associated antiviral mechanism in depth. HAX1 hijacks the viral proteins (especially the driver of virion packaging, Gn) to mitochondria and interferes with the Golgi localization, inhibiting virion packaging and propagation. Thus, HAX1 itself could be considered as an example regulating the viral packaging/assembly. Indeed, based on the interactomes established here, we have also found interesting candidates probably participating in other aspects including the viral entry and the work is still being explored in the lab. Thus, together with the detailed exemplification of HAX1, we consider that the study is informative and could be of importance and interest to the field. Thanks for giving us to further discuss and explain the points. Again, we are very grateful to the reviewer for all these important and helpful comments and suggestions.

REVIEWERS' COMMENTS

Reviewer #1 (Remarks to the Author):

The authors have provided all the changes required and new data to support these changes. No further revision is required from my perspective.

Reviewer #2 (Remarks to the Author):

I have no further comments to add to this manuscript. The authors addressed all the suggestions diligently and performed additional experiments and analysis to further support their findings. This revised version is greatly improved.

Reviewer #3 (Remarks to the Author):

Thank you very much for addressing all the comments, for performing additional experiments, and revising the manuscript.